# AGENTIC ANALOGICAL REASONING FOR LARGE LANGUAGE MODELS

## ABSTRACT

Analogical reasoning helps humans grasp new concepts by relating them to familiar ones. Recent work seeks to improve LLM reasoning by prompting analogical correspondences with semantically related scenarios. However, existing approaches are single-turn reasoning and may generate unreliable analogical instances, which restricts their effectiveness in complex reasoning tasks. To address these limitations, we propose a novel **A**gentic **A**nalogical **R**easoning (**AAR**) paradigm for LLM reasoning. This paradigm treats the LLM as an agentic reasoner to integrate multi-turn insights along the reasoning trajectory of iteratively generating analogical queries to trigger internal or external knowledge for analogical exemplification, and selectively identifying appropriate analogies to conduct further reasoning. To equip LLMs with AAR capability, we design an analogical trajectory optimization algorithm including analogical trajectory generation and re-weighted trajectory training. Furthermore, a mixed training strategy is devised to progressively internalize agentic analogical reasoning as an intrinsic capability of LLMs. Finally, we conduct extensive experiments on seven reasoning-intensive datasets and achieve significant performance improvements over prior state-of-the-art (SOTA) methods. The code is available at `https://anonymous.4open.science/r/ICLR-8381`.

## 1 INTRODUCTION

Large Language Models (LLMs) have demonstrated remarkable proficiency across diverse reasoning tasks, advancing from intuitive "fast thinking" to more deliberate "slow thinking" (Wang et al., 2022a; Chen et al., 2024). This evolution in cognitive processing has enabled LLMs to exhibit human-like cognition capabilities such as self-verification and multi-step reasoning (Kumar et al., 2024; Wu et al., 2024). Despite recent advances, LLMs still fall short of many key aspects of human cognition such as analogical reasoning, a crucial mechanism for solving complex and creative problems by mapping unfamiliar situations onto familiar domains and transferring insights across contexts (Gentner, 1983; Hofstadter & Sander, 2013). Empowering LLMs with analogical reasoning can allow them to efficiently migrate similar knowledge across different problems to strengthen their reasoning capacity.

Early studies on analogical reasoning in LLMs primarily examined whether models exhibit analogical capabilities, typically through test cases constructed from text pairs with lexical or structural similarity(Chen et al., 2022; Czinczoll et al., 2022). Recently, some research has shifted toward leveraging the mechanism of analogy to enhance the reasoning capabilities of LLMs. These methods utilize the LLMs' internal knowledge to recall semantically related scenarios and establish analogical correspondences as reasoning guidance (Yu et al., 2023; Feng et al., 2024). For instance, Thought Propagation (Yu et al., 2023) prompts LLMs to construct and solve a set of analogous problems related to the original input, and then reuses the solutions either to directly generate an answer or to formulate a knowledge-intensive plan that refines the initial solution. Analogical Prompting (Feng et al., 2024) further requires LLMs to autonomously generate tutorial knowledge in addition to task-specific exemplars, in order to provide a more comprehensive background for analogical reasoning.

However, prompting LLMs to directly generate answers through analogical reasoning still faces limitations. First, existing approaches struggle with complex reasoning tasks with large spans. Their reasoning processes are typically shallow and single-turn, grounded in direct inference from one or more analogies, and therefore lack the progressive, multi-turn capacity these tasks demand. Second, the exclusive reliance on the internal knowledge of LLMs can lead to hallucinations, producing

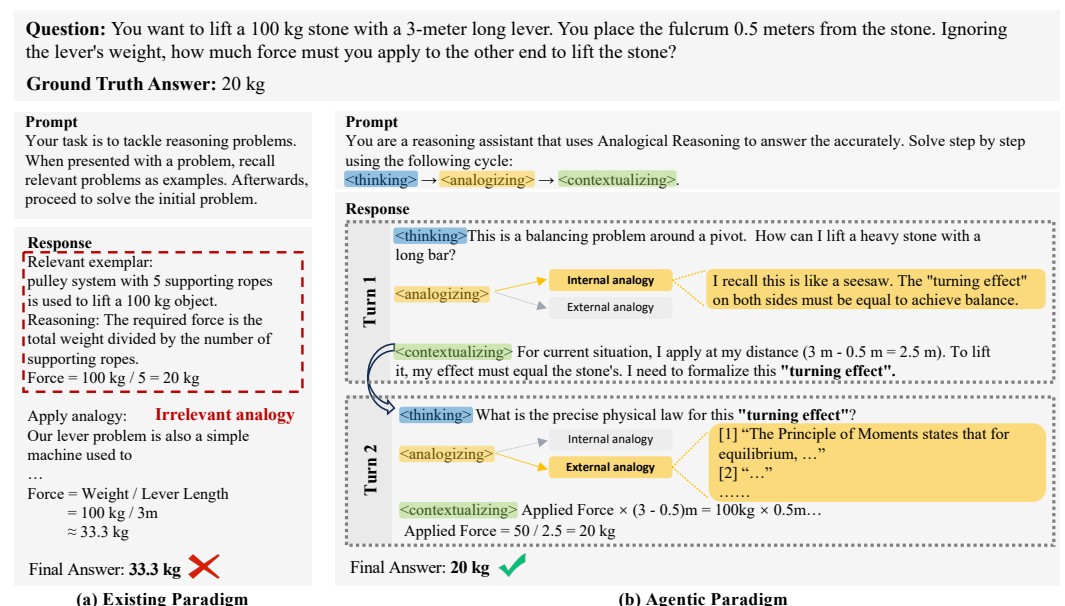

Figure 1: Comparison of different analogical reasoning paradigms.

fictitious or factually incorrect analogical instances that undermine the credibility and reliability of the reasoning process. Third, the effectiveness of reasoning is highly contingent on the quality of generated analogical instances, while current methods lack mechanisms to reliably identify appropriate analogies, making it easy for errors to propagate through the reasoning process.

To address these limitations, we propose a novel **A**gentic **A**nalogical **R**easoning (**AAR**) paradigm which treats the LLM as an agentic reasoner to integrate multi-turn insights along the reasoning trajectory of iteratively generating analogical queries to trigger internal or external knowledge for analogical exemplification, and selectively identifying appropriate analogy to conduct further reasoning, as exemplified in Figure 1. Specifically, we formalize an agentic analogical reasoning trajectory composed of multiple continuous analogical units. In each turn, we execute three actions as an analogical unit: (1) *thinking*, generating analogical queries adaptively based on the current reasoning state; (2) *analogizing*, triggering analogical knowledge instances from internal or external knowledge; and (3) *contextualizing*, filtering appropriate analogy to derive insights from analogical scenarios for the current context. To equip LLMs with AAR capability, we design an analogical trajectory optimization algorithm. We first trigger external knowledge analogy to generate reasoning trajectories and leverage the LLM to assign importance weights to each trajectory based on its support to correct answer. Then, we integrate trajectory reweighting into the ELBO objective function to encourage the generation of more supportive trajectories to obtain accurate answers. Furthermore, we devise a mixed training strategy that progressively enhances the intrinsic analogical capabilities of LLMs by leveraging both self-generated and external-retrieved analogical trajectories. We conduct extensive experiments on seven reasoning-intensive datasets and achieve significant performance improvements over previous SOTA methods. To summarize, our main contributions are the following: (1) We propose a novel Agentic Analogical Reasoning (AAR) paradigm which treats the LLM as an agentic reasoner to integrate multi-turn insights along the reasoning trajectory. (2) We specially design an analogical trajectory optimization algorithm to equip LLMs with AAR capability. (3) We devise a mixed training strategy to progressively internalize agentic analogical reasoning into LLMs. (4) We conduct extensive experiments on seven reasoning-intensive benchmarks, demonstrating the superiority of the proposed AAR.

## 2 RELATED WORKS

**LLM Reasoning** Large Language Models (LLMs) have demonstrated remarkable reasoning capabilities, including reflection, backtracking, and multi-path thinking (Wang et al., 2022a; Chen et al., 2024; Kumar et al., 2024; Wu et al., 2024; Ma et al., 2025), marking a shift from simply answering questions

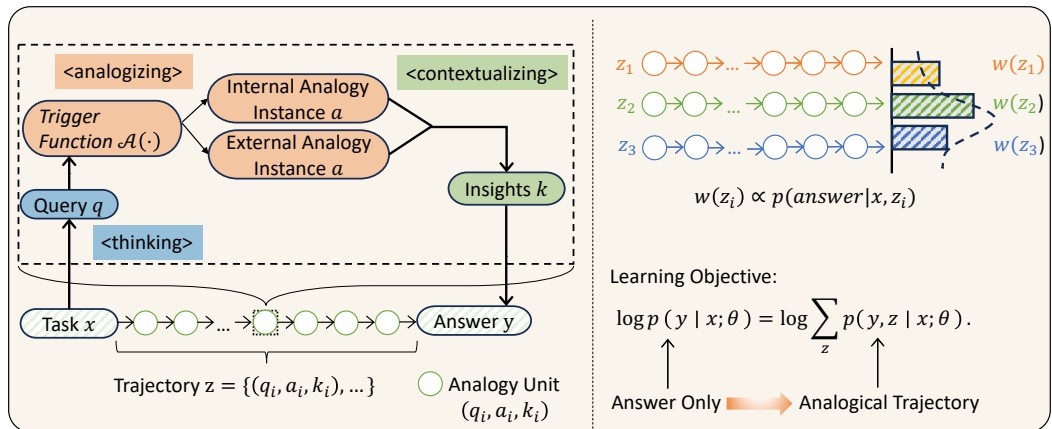

Figure 2: Illustration of Agentic Analogical Reasoning (AAR).

to exhibiting higher-order cognitive traits (Chen et al., 2025a). Recent work has further explored enhancing such abilities, for example by training inductive–deductive skills to improve performance on complex scientific problems (Hu et al., 2025). In parallel, a growing line of research focuses on analogical reasoning—drawing from relevant past experiences to tackle new problems—drawing inspiration from classical cognitive theories such as structure-mapping (Gentner, 1983; Hofstadter & Sander, 2013). Early computational studies addressed lexical or relational analogies (Chen et al., 2022; Czinczoll et al., 2022), while more recent efforts extend to analogies across systems, processes, paragraphs, and stories (Yuan et al., 2023; Bhavya et al., 2022; Webb et al., 2022; Jiayang et al., 2023). Methods such as Analogical Prompting (Yasunaga et al., 2023b), Meta-Ability Alignment (Hu et al., 2025), Thought Propagation (Yu et al., 2023), and Thought Retriever (Feng et al., 2024) attempt to integrate analogy generation into reasoning chains. Yet, these approaches often remain surface-level, hindered by noise accumulation or reliance on external knowledge bases, leaving the development of a general and robust framework for deep cross-contextual reasoning an open challenge.

**Retrieval-Augmented Language Models** The retrieval-augmented (Large) Language Model has been proposed as a solution to alleviate hallucinations and improve the output quality of language models (Asai et al., 2023; Mialon et al., 2023; Shi et al., 2023; Gao et al., 2023; Zhao et al., 2024). When faced with an input query, a retrieval-augmented LLM first searches an external database containing billions of tokens (Borgeaud et al., 2022) to obtain a subset of textual corpus that assists in generating the final answer. Notably, retrieval-augmented LLMs achieve superior performance while using significantly fewer parameters compared to conventional LLMs (Mialon et al., 2023), and have already been applied to various downstream tasks (Yasunaga et al., 2023a; Izacard et al., 2023; Wang et al., 2022b), including multimodal generation (Yasunaga et al., 2023a; Zhao et al., 2024; Borgeaud et al., 2022; Izacard et al., 2023) as well as biomedical applications (Wang et al., 2022b; Yang et al., 2023). Recently, ExSearch (Shi et al., 2025) concentrated on refining multi-turn retrieval mechanisms to search for factual evidence from external knowledge. However, this paradigm is insufficient for complex reasoning tasks, which require reasoning capabilities instead of merely relying on retrieval. To address this, our approach diverges by enhancing LLMs with analogy reasoning ability.

## 3 METHODOLOGY

In this section, we first introduce how the proposed method **A**gentic **A**nalogical **R**easoning (AAR), models reasoning as an iterative construction of analogy trajectories, interleaving the three core actions of thinking, analogizing, and contextualizing. Next, we describe how these trajectories are optimized to improve reasoning performance. Finally, we present the process of analogical capability internalization, where enhanced trajectories progressively guide the model from external retrieval toward autonomous internal analogy generation. Figure 2 presents the overall framework.

## 3.1 ANALOGICAL REASONING TRAJECTORY

We draw inspiration from structure-mapping theory (Gentner, 1983), a foundational cognitive paradigm that conceptualizes analogical reasoning as a dynamic process of mapping a source domain onto a target domain. Accordingly, AAR simulates this process by interleaving three core agentic actions: *thinking*, *analogizing*, and *contextualizing*. These three actions collectively form an **analogical unit**, the fundamental building block that progressively constructs the overall reasoning trajectory. Taking the $i$-turn as an example, we describe three actions within the analogical unit.

**Thinking.** Given an input task $x$ and the current reasoning context, the model first generates an analogy query $q_i$. This action allows the model to proactively identify aspects of the problem that require analogical support, thereby outlining a coarse-grained analogical domain for the subsequent analogy.

**Analogizing.** Based on the query $q_i$, this action triggers analogical instances $a_i$ to provide support. The relevance and accuracy of analogies are critical, as inaccurate cases can lead to flawed reasoning (Qin et al., 2025). To address this, we empower the model with the analogy ability to draw from both *internal* and *external* sources. Formally, we define a trigger function $\mathcal{A}(\cdot)$ that adaptively selects between these two sources:

- **External Analogy**: When the model emits a special token (e.g., "<analogizing>"), the function $\mathcal{A}$ invokes an external retrieval module to search a knowledge base containing factual and reasoning knowledge, returning relevant documents as the analogy $a_i$.
- **Internal Analogy**: In other cases, the model generates the analogy from its own parametric knowledge, often prefaced with phrases like "I recall a similar problem...".

**Contextualizing.** This action mirrors the core mapping process in structure-mapping theory. It grounds the triggered analogy $a_i$ within the current reasoning context, extracting key insights $\kappa_i$. This step is crucial for filtering out irrelevant information from the analogical instance and ensuring the reasoning process remains on track, preventing potential deviations.

An analogical unit thus completes a full cycle of $(q_i, a_i, \kappa_i)$. To tackle complex problems that require progressive deepening of thought, the model executes multiple analogical units in sequence. The entire reasoning process is captured by an **analogical trajectory $\boldsymbol{z}$**, which is a sequence of these analogical units: $\boldsymbol{z} = \{(q_i, a_i, \kappa_i)\}_{i=1}^{|\boldsymbol{z}|}$. The likelihood of this trajectory is given by:

$$p(\boldsymbol{z} \mid x; \theta) = \prod_{i=1}^{|\boldsymbol{z}|} p\left((q_i, a_i, \kappa_i)|x, \boldsymbol{z}_{<i}; \theta\right) = \prod_{i=1}^{|\boldsymbol{z}|} p\left(q_i \mid x, \boldsymbol{z}_{<i}; \theta\right) \cdot p\left(\kappa_i \mid q_i, \mathcal{A}(q_i); \theta\right), \quad (1)$$

where $\theta$ denotes the LLM parameters and $x$ is the input task. After this iterative process, the final answer $y$ is generated, denoted as: $y \sim p(y \mid x, \boldsymbol{z}; \theta)$. The system prompt used in AAR is provided in Appendix A.2.

## 3.2 ANALOGICAL TRAJECTORY OPTIMIZATION

Applying the AAR framework to analogical reasoning is fundamentally about optimizing the final answer $y$. In this section, we describe how our learning objective transforms the optimization of $y$ into an optimization related to the analogical trajectory $\boldsymbol{z}$. Adopting a variational inference framework (Kingma & Welling, 2013), the basic procedure is as follows: we treat the analogical reasoning trajectory $z$ as a latent variable. Since directly inferring the target posterior distribution $p(z|x, y)$ is intractable, the model is trained by maximizing the ELBO (Evidence Lower Bound), which is derived via Jensen's inequality. During optimization, we utilize Importance Sampling, using the prior distribution $p(z|x)$ as the sampling distribution to approximate the target posterior distribution $p(z|x, y)$. This is consistent with the PPO (Schulman et al., 2017) in reinforcement learning, where samples from an old policy (analogous to the prior) are used to optimize the new policy. From a statistical perspective, this corresponds to the Generalized Expectation-Maximization (GEM) algorithm (Dempster et al., 1977). The details are as follows.

**Estimation of the Learning Objective.** Given the input task $x$, we define the training objective of AAR as optimizing the ability of LLMs to generate the correct answer $y$ after the reasoning trajectory $\boldsymbol{z}$. This can be modeled as the joint probability of the generation and reasoning processes:

$$\log p(y \mid x; \theta) = \log \sum_{\boldsymbol{z}} p(y, \boldsymbol{z} \mid x; \theta). \quad (2)$$

Here, $\boldsymbol{z} = \{(q_i, a_i, \kappa_i) \mid i \in [|\boldsymbol{z}|]\}$ represents an analogical reasoning trajectory as introduced in 3.1. However, since marginalizing over the vast space of all possible trajectories $\boldsymbol{z}$ is intractable, we treat $\boldsymbol{z}$ as a latent variable and derive a tractable objective using variational inference. To do this, we first introduce an auxiliary proposal distribution $q(\boldsymbol{z} \mid x)$, which serves as a manageable approximation from which we can sample. Then, by applying *Jensen's inequality* which creates a lower bound for the log of a sum, we transform the marginal log-likelihood in Eq. 2:

$$\log \sum_{\boldsymbol{z}} q(\boldsymbol{z}|x) \frac{p(y, \boldsymbol{z}|x; \theta)}{q(\boldsymbol{z}|x)} \geq \sum_{\boldsymbol{z}} q(\boldsymbol{z}|x) \log \frac{p(y, \boldsymbol{z}|x; \theta)}{q(\boldsymbol{z}|x)} = \mathbb{E}_{\boldsymbol{z} \sim q(\boldsymbol{z}|x)} \left[ \log \frac{p(y, \boldsymbol{z}|x; \theta)}{q(\boldsymbol{z}|x)} \right]. \quad (3)$$

We refer to this expectation $\mathbb{E}_{\boldsymbol{z} \sim q(\boldsymbol{z}|x)}[\log \frac{p(y, \boldsymbol{z}|x; \theta)}{q(\boldsymbol{z}|x)}]$ as the variational evidence lower bound (ELBO) of $\log p(y|x; \theta)$, which provides a tractable objective for optimization. The following parts detail the optimization process based on this objective, starting from the sampling trajectory generation.

**Analogical Trajectory Generation.** In the $t$-th training iteration, to ensure the boundedness of Eq. 3, we assume $q(z \mid x) \approx p(z \mid x, y; \theta^t)$ and rewrite the ELBO as follows. The $\mathcal{H}(\cdot)$ denotes the entropy.

$$\mathbb{E}_{\boldsymbol{z} \sim p(\boldsymbol{z}|x, y; \theta^t)} \left[\log p(y, \boldsymbol{z} \mid x; \theta)\right] + \mathcal{H}(p(\boldsymbol{z} \mid x, y; \theta^t)). \quad (4)$$

The entropy $\mathcal{H}(p(z \mid x, y; \theta^t))$ is independent of the model parameters $\theta$ and can be treated as a constant and ignored. To make the expectation in Eq. 4 tractable, we employ an *importance sampling* strategy Elvira & Martino (2021); Tokdar & Kass (2010). This allows us to sample from the simpler prior distribution $p(z \mid x; \theta^t)$ instead of the complex posterior $p(z \mid x, y; \theta^t)$. The ELBO can then be rewritten as: $\mathbb{E}_{\boldsymbol{z} \sim p(\boldsymbol{z}|x; \theta^t)}[w(\boldsymbol{z}) \log p(y, \boldsymbol{z} \mid x; \theta)] + c$, where the importance weight is defined as $w(z) := \frac{p(z|x, y; \theta^t)}{p(z|x; \theta^t)}$. By the product rule of probability, the posterior in the numerator can be expressed as $p(z \mid x, y; \theta^t) \propto p(z \mid x; \theta^t) \times p(y \mid x, z; \theta^t)$, the weight simplifies to $w(z) \propto p(y \mid z, x; \theta^t)$. This weight directly reflects how much the analogical trajectory $\boldsymbol{z}$ supports the correct answer $y$.

**Training with Re-weighted Trajectories.** We then update the model parameters $\theta$ by maximizing the ELBO. By applying the product rule to the joint likelihood $p(y, z \mid x; \theta)$, this objective can be decomposed as: $\theta = \arg\max_\theta \mathbb{E}_{z \sim p(z|x; \theta^t)}[w(z) \log p(z \mid x; \theta) + w(z) \log p(y \mid x, z; \theta)]$.

This decomposition reveals two complementary roles in training: (i) the term $w(z) \log p(z \mid x; \theta)$ encourages the model to generate high-quality analogical trajectories, while (ii) the term $w(z) \log p(y \mid x, z; \theta)$ guides the model to integrate information from these trajectories to produce the correct answer. This decomposition provides a lens through which the training process can be interpreted from the dual aspects of analogical trajectory generation and answer aggregation. The overall optimization in this stage can be achieved via stochastic gradient descent, and the gradient with respect to $\theta$ is:

$$\nabla_\theta \text{ELBO}(\theta) = -\mathbb{E}_{z \sim p(z|x; \theta^t)} \left[ w(z) \nabla_\theta \big( \log p(z \mid x; \theta) + \log p(y \mid x, z; \theta) \big) \right]. \quad (5)$$

More details about the practical implementation of the optimization are presented in Appendix A.7.

### 3.3 Analogical Capability Internalization

The agentic analogical training operates through an iterative mechanism designed to progressively internalize analogical capabilities. This process unfolds across training iterations, beginning with a preliminary phase lasting for $N_e$ iterations. During this phase, the trigger function $a_i = \mathcal{A}(q_i)$ is implemented primarily via external retrieval, sourcing relevant instance knowledge to build a foundation of high-quality analogical patterns. The model is trained throughout this phase using the analogical training algorithm described in Section 3.2.

Following the preliminary phase, the training process focuses on gradually enhancing the model's intrinsic analogical abilities. This is achieved by enriching the existing training data with "enhanced" trajectories. Specifically, we construct these trajectories by reframing the external analogy steps into self-generated, internalized analogies. For example, a factual, retrieved analogy such as, "An analogous problem is the Tower of Hanoi," is transformed into a reflective pattern like, "I recall a similar problem, the Tower of Hanoi..." (the full prompt for this transformation is detailed in Appendix A.3). By integrating these enhanced trajectories for the following iterations, the model continues to be optimized using the same agentic analogical training algorithm from Section 3.2. This iterative refinement progressively guides the trigger function $\mathcal{A}$ with internal generation, further stimulating the model's intrinsic capability for autonomous analogical reasoning.

## 4 EXPERIMENTS

### 4.1 DATASETS AND METRICS

Our experimental setting draws inspiration from relevant works (Li et al., 2025; Jin et al., 2025; Shi et al., 2025). We evaluate our proposed AAR on a comprehensive suite of benchmarks spanning two categories. **General multi-hop reasoning**: (1) **Natural Questions (NQ)** Kwiatkowski et al. (2019), containing real-world questions from Google search queries; (2) **HotpotQA** (Yang et al., 2018), requiring reasoning over multiple Wikipedia passages; (3) **MuSiQue** (Trivedi et al., 2022), designed for multi-step reasoning with compositional questions; and (4) **2WikiMultihopQA (2WikiQA)** (Ho et al., 2020), featuring multi-hop reasoning across Wikipedia entities. **Scientific reasoning**: (1) **AIME2024** (AI-MO, 2024a), problems from the American Invitational Mathematics Examination requiring advanced mathematical reasoning; (2) **AMC2023** (AI-MO, 2024b), middle school mathematics competition problems covering arithmetic, algebra, and geometry; (3) **GPQA** (Rein et al., 2024), a PhD-level science multiple-choice dataset with questions authored by domain experts in physics, chemistry, and biology. In our experiments, we use three metrics to measure the model performance: F1 score, Exact Match (EM), and Accuracy (Acc).

### 4.2 BASELINES

We compare AAR with a range of baselines, categorized into three groups based on their reasoning strategies: (1) **Direct Reasoning**: These methods produce the answer to the input query by prompting the LLM to reason over its parametric internal knowledge. This includes few-shot prompting off-the-shelf LLMs, i.e., DeepSeek-R1 Guo et al. (2025), GPT-4o OpenAI (2023), GPT-3.5 OpenAI (2022), Qwen2.5 Yang et al. (2024), QwQ-32B Team (2024), LLaMA-3.3-70B (Dubey et al., 2024), and Mistral-8x7B Jiang et al. (2023). (2) **Retrieval-Augmented Generation**: These methods incorporate external knowledge retrieval to support the reasoning process. We include both advanced RAG methods that retrieve relevant documents, followed by filtering or re-ranking: InstructRAG Wei et al. (2025), RetRobust Yoran et al. (2023), RAG-DDR Li et al. (2024) (trained by DPO Rafailov et al. (2024)), and ChatQA Liu et al. (2024); as well as multi-turn methods that allow LLMs to interact with the retriever through multiple rounds: GenGround Shi et al. (2024), DSPy Khattab et al. (2024), Iter-RetGen Shao et al. (2023), Gen-Ret-Gen Abdallah & Jatowt (2023), Search-o1 Li et al. (2025), Search-R1 Jin et al. (2025) (trained by PPO Schulman et al. (2017)), Self-RAG Asai et al. (2024) and ExSearch (Shi et al., 2025). (3) **Analogical Reasoning**: These methods explicitly enhance reasoning through analogical mechanisms, which directly align with our motivation. We include *Analogical Prompting* Yasunaga et al. (2023b), which prompts models to recall and apply analogical cases, and *Meta-Ability Alignment* Hu et al. (2025), which augments cognitive ability of LLMs explicitly. More details are presented in Appendix A.5.

#### 4.2.1 IMPLEMENTATION DETAILS

For training, we merge the training sets of NQ and HotpotQA to form a unified dataset following (Jin et al., 2025). Evaluation is conducted on the test sets of four datasets to assess both in-domain and out-of-domain performance. We implement AAR on Qwen-2.5-7B-Instruct and Llama-3.1-8B. Following prior work (Xu et al., 2024), we build by combining the December 20, 2018 Wikipedia passage dump with reasoning trajectories from ReasonFlux (Yang et al., 2025) and OpenThoughts (Guha et al., 2025). We adopt ColBERTv2.0 (Santhanam et al., 2021) for document retrieval. To equip the model with the basic ability to follow the "thinking–analogizing–contextualizing" routine, we first synthesize 1,000 pseudo-examples and perform a small-scale fine-tuning under a standard cold-start setup, following a cold-start setup as in prior work (Chen et al., 2025b; Song et al., 2025). For subsequent trajectory generation, we set the number of iterations $N=5$ and report model performance at each iteration. The preliminary phase iteration $N_e$ is set as 2. Appendix A.10 demonstrates the versatility of AAR, reporting consistent gains across diverse retrievers like E5 and DPR.

### 4.3 EXPERIMENT RESULTS

**Effectiveness on General Multi-hop QA Tasks.** As shown in Table 1, our method demonstrates substantial improvements across all general reasoning benchmarks. Compared to direct reasoning methods, despite using a smaller backbone model, our Qwen-2.5-7B based approach achieves a 9.9% relative F1 improvement over GPT-4o, demonstrating that AAR can more effectively understand

Table 1: Comparison between our proposed AAR and baselines, with the **best** and **second best** results in bold. * indicates cases where the model produces long-form answers, which struggle to align with the short-span ground truth format. In these cases, we suggest using Acc. as a more reliable metric.

| Tasks | NQ | | | HotpotQA | | | MuSiQue | | | 2WikiQA | | | Avg. | | |
|---|---|---|---|---|---|---|---|---|---|---|---|---|---|---|---|
| Metrics | F1 | EM | Acc. | F1 | EM | Acc. | F1 | EM | Acc. | F1 | EM | Acc. | F1 | EM | Acc. |
| *Direct Reasoning Methods* | | | | | | | | | | | | | | | |
| Mistral-8x7B | 40.87 | 40.10 | 39.60 | 25.19 | 16.40 | 25.80 | 11.60 | 6.80 | 14.80 | 30.21 | 27.05 | 29.67 | 26.97 | 22.59 | 27.47 |
| Qwen-2.5-32B | 33.09 | 23.39 | 32.20 | 33.34 | 25.40 | 26.60 | 18.85 | 9.00 | 9.00 | 40.90 | 34.80 | 36.40 | 31.55 | 22.95 | 26.05 |
| Qwen-2.5-70B | 48.70 | 36.01 | 36.00 | 48.10 | 37.80 | 39.20 | 23.60 | 14.80 | 22.64 | 54.20 | 46.00 | 50.00 | 43.90 | 33.68 | 35.46 |
| QwQ-32B | 33.10 | 23.40 | 34.20 | 34.74 | 25.40 | 27.60 | 18.90 | 8.50 | 10.03 | 36.34 | 29.40 | 34.40 | 30.77 | 21.68 | 26.58 |
| Llama-3.3-70B | 45.68 | 30.12 | 45.67 | 39.80 | 29.20 | 32.00 | 20.40 | 11.40 | 15.00 | 42.70 | 34.40 | 33.61 | 36.90 | 26.25 | 30.95 |
| Deepseek-R1-671B | 49.45 | 35.71 | 43.83 | 46.98 | 35.83 | 37.80 | 17.34 | 10.22 | 12.69 | 52.18 | 43.83 | 50.66 | 41.49 | 31.40 | 36.24 |
| GPT-4o | 48.76 | 35.75 | 43.03 | 54.13 | 36.52 | 51.59 | 29.07 | 18.92 | 22.97 | 51.31 | 40.45 | 53.07 | 45.82 | 32.90 | 42.66 |
| GPT-3.5-turbo | 42.11 | 38.60 | 40.60 | 34.90 | 24.57 | 31.86 | 22.73 | 14.14 | 16.29 | 33.90 | 30.40 | 32.45 | 33.41 | 26.93 | 30.30 |
| *Retrieval-Augmented Generation Methods* | | | | | | | | | | | | | | | |
| InstructRAG (Llama-3.1-8B) | 43.21 | 37.03 | 39.56 | 40.54 | 35.59 | 38.39 | 18.16 | 18.11 | 19.61 | 39.11 | 38.65 | 39.97 | 35.61 | 32.32 | 34.36 |
| RetroRust (Llama-2-13B) | 39.82 | 27.82 | 37.58 | 37.31 | 36.77 | 35.71 | 25.88 | 14.94 | 20.45 | 40.01 | 44.57 | 38.71 | 35.40 | 33.54 | 33.04 |
| RAG-DDR (Llama-3.1-8B) | 40.74 | 28.76 | 30.51 | 31.71 | 40.44 | 42.41 | 13.54 | 10.57 | 14.21 | 38.40 | 35.44 | 37.41 | 31.10 | 28.70 | 31.14 |
| ChatQA (Llama-2-7B) | 34.54 | 23.64 | 37.41 | 44.60 | 33.40 | 33.40 | 17.05 | 16.64 | 19.24 | 31.90 | 26.80 | 32.56 | 32.02 | 25.12 | 30.65 |
| GenGround (GPT-3.5) | 50.31 | 40.24 | 43.60 | 52.26 | 45.31 | 47.27 | 27.36 | 18.34 | 20.24 | 50.21 | 42.31 | 45.61 | 45.04 | 36.55 | 39.18 |
| IRCoT (GPT-3.5) | 45.42 | 42.41 | 43.21 | 58.40 | 45.50 | 46.32 | 30.50 | 19.01 | 22.87 | 45.10 | 35.40 | 36.54 | 44.86 | 35.58 | 37.24 |
| DSPy (GPT-3.5) | 42.25 | 29.10 | 42.00 | 47.10 | 34.67 | 42.73 | 19.88 | 10.80 | 13.40 | 44.52 | 39.64 | 44.43 | 38.44 | 28.55 | 35.64 |
| Iter-RetGen (GPT-3.5) | 28.30 | – | 41.04 | 44.10 | – | 21.04 | 17.69 | – | 20.19 | 36.00 | – | 42.17 | 31.52 | – | 31.11 |
| Gen-Ret-Gen (GPT-3.5) | 46.66 | 38.08 | 48.38 | 49.59 | 37.69 | 45.43 | 25.94 | 13.24 | 17.81 | 40.26 | 29.43 | 39.63 | 40.61 | 29.60 | 37.62 |
| Search-o1 (QwQ-32B) | 47.52 | 32.41 | 40.34 | 53.31 | 43.51 | 45.31 | 25.41 | 16.64 | 19.42 | 50.31 | 42.61 | 45.41 | 44.14 | 33.79 | 37.62 |
| Search-R1 (Qwen-2.5-7B) | 53.26 | 42.21 | 49.75 | 58.04 | 46.51 | 52.46 | 26.51 | 19.60 | 22.10 | 52.62 | 49.64 | 50.43 | 48.01 | 39.89 | 44.90 |
| SELF-RAG (Qwen-2.5-7B) | 49.70 | 41.60 | 42.50 | 21.50 | 9.40 | 29.20 | 20.50 | 9.43 | 7.10 | 27.33 | 23.52 | 20.80 | 30.74 | 20.99 | 24.02 |
| ExSearch(Qwen-2.5-7B) | **55.43** | **46.81** | **50.83** | 60.96 | 49.36 | 54.01 | 27.47 | 20.97 | 23.16 | 57.08 | 52.46 | 54.26 | 50.24 | 42.40 | 45.57 |
| *Analogical Reasoning Methods* | | | | | | | | | | | | | | | |
| Analogical Prompting (GPT-4o) | 45.15 | 32.60 | 41.83 | 53.57 | 35.13 | 50.26 | 28.15 | 17.12 | 21.34 | 50.12 | 39.17 | 51.78 | 44.23 | 30.16 | 40.97 |
| Meta-Ability Alignment (Qwen-2.5-7B) | 47.15 | 40.23 | 43.87 | 50.64 | 38.91 | 44.26 | 25.13 | 17.45 | 20.12 | 45.82 | 38.73 | 43.29 | 42.19 | 33.83 | 37.89 |
| **AAR-Qwen-2.5-7B** | 55.37 | 46.70 | 50.75 | **61.05** | **49.65** | **54.13** | 27.68 | **21.03** | **23.34** | 57.31 | 52.82 | 54.43 | **50.35** | **42.55** | **45.66** |
| **AAR-Llama-3.1-8B** | 54.21 | 43.70 | 50.20 | 60.81 | 47.58 | 53.26 | 26.78 | 20.54 | 22.71 | 54.31 | 46.21 | 54.34 | 49.03 | 39.51 | 45.13 |

complex problems beyond parametric knowledge. This advantage stems from AAR to not only retrieve factual knowledge, but more importantly, establish iterative analogical associations during the reasoning process, achieving dynamic optimization of reasoning paths and deep integration of knowledge through the cyclical "think-analogize-contextualize" mechanism. Most importantly, our method significantly surpasses existing analogical reasoning baselines, achieving a 20.6% relative accuracy improvement over Meta-Ability Alignment. Unlike prompt-based analogical methods, our iterative trajectory learning mechanism can systematically train the model to learn and interact with analogical examples, rather than relying on static analogical prompts, thus achieving more continuous reasoning between analogical cases and current reasoning. Moreover, our analysis of the computational budget in Appendix A.8 indicates that AAR demonstrates favorable efficiency in terms of training and inference costs.

**Effectiveness on Mathematical and Scientific Reasoning.** Table 2 show the results of performance comparison on scientific reasoning tasks. From the overview, our method achieve pronounced improvements. Note that some retrieval-based baselines specifically trained for retrieval augmentation (e.g., RAG-DDR, InstructRAG) show suboptimal performance on these specialized domains due to insufficient training

Table 2: Results on scientific reasoning benchmarks.

| Method | GPQA (PhD-Level Science QA) | | | | Math | |
|---|---|---|---|---|---|---|
| | Physics | Chemistry | Biology | Overall | AIME24 | AMC23 |
| Qwen2.5-7B-Instruct | 28.12 | 22.47 | 24.35 | 24.60 | 15.00 | 53.00 |
| Chain-of-Thoughts | 30.28 | 25.19 | 26.41 | 27.00 | 18.00 | 55.00 |
| ChatQA | 31.85 | 25.90 | 28.73 | 28.83 | 20.10 | 61.50 |
| Search-o1 | 36.05 | 29.60 | 33.90 | 32.10 | 31.20 | 73.10 |
| Analogy-Prompting | 27.36 | 20.42 | 22.18 | 21.00 | 10.00 | 50.00 |
| Meta-ability-Argumented | 36.08 | 28.27 | 30.44 | 33.50 | 22.90 | 52.30 |
| **AAR-Qwen2.5-7B** | **40.25** | **33.18** | **37.42** | **36.95** | **33.50** | **76.20** |
| **AAR-Llama3.1-8B** | **39.18** | **31.85** | **35.73** | **35.59** | **35.20** | **74.80** |

data coverage in mathematics and sciences, and thus are not reported in this table. In GPQA, we reach 36.95% overall accuracy, outperforming the strongest RAG baseline Search-R1 by +4.45 percentage points, with particularly strong performance in Physics (40.25%) and Biology (37.42%). For mathematical reasoning, we achieve 33.50% on AIME2024, representing a +10.6 percentage point improvement over the strongest analogical reasoning baseline Meta-Ability Alignment (22.90%), and 76.20% on AMC2023, improving by +23.9 percentage points over 52.30%. This significant improvement in specialized domains stems from our iterative analogical mechanism's ability to better identify and utilize cross-problem knowledge patterns and reasoning strategies: through the analogy process, the model can effectively utilize solution patterns from similar problems. Unlike general

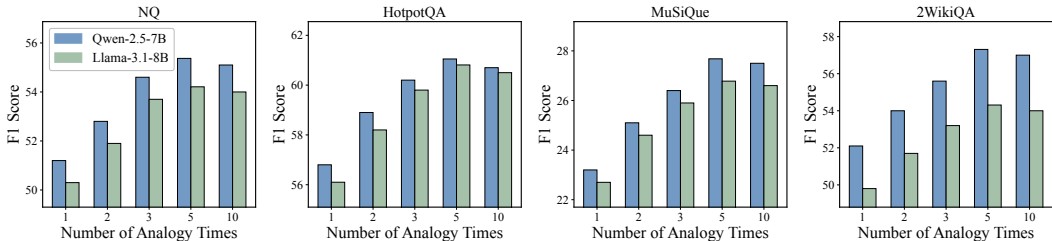

Figure 3: Scaling analysis on the number of analogical reasoning steps. We report F1 scores on four multi-hop QA datasets using Qwen-2.5-7B and Llama-3.1-8B backbones. Performance improves with more steps up to 5, after which gains saturate or slightly decline.

reasoning tasks, these specialized domain tasks rely more heavily on deep conceptual understanding and pattern transfer capabilities, which are core strengths of our analogical reasoning framework. Consistent improvements across both mathematical and scientific reasoning validate the potential applicability for challenging of the AAR framework.

## 4.4 ABLATION STUDY

We validate the effectiveness of our five core components by independently removing each component and evaluating the resulting variants. As shown in Table 3, the results are summarized as follows: (1) **w/o thinking**: Removing thinking reduces our method to a standard question-then-generate pipeline, leading to a substantial performance drop across all datasets. This validates the importance of adaptive trigger point identification, as without it the model degenerates into a one-turn pipeline without iterative refinement. (2) **w/o analogizing**: Without analogizing, the method cannot leverage external analogical knowledge retrieval and must rely solely on the model's internal parametric knowledge for analogical reasoning. We observe a consistent decline across all datasets, validating the necessity of analogizing from external knowledge access mechanism, limiting in internal analogy of LLMs cannot leverage diverse external experiences. (3) **w/o contextualizing**: Disabling contextualizing prevents the model from extracting relevant insights for each analogical subquery, leading to 6% drop. This suggests contextualizing is crucial for converting abstract analogies into actionable insights, as otherwise analogies remain at the surface level without guiding reasoning. (4) **w/o learning stage**: Removing the learning stage reduces our method to a naive training setup where all reasoning paths are treated equally, showing drops from 61.05 to 57.26. This highlights the role of trajectory quality discrimination, as the learning stage teaches the model to prioritize effective reasoning patterns via weighted optimization. (5) **w/o internalizing**: Removing internalizing causes a modest but consistent drop, indicating internalizing remains important for progressively reducing reliance on retrieval and fostering autonomous analogical reasoning.

Table 3: Ablation study where we remove each component from our method.

| Method | HotpotQA | | | 2WikiQA | | |
|---|---|---|---|---|---|---|
| | F1 | EM | Acc. | F1 | EM | Acc. |
| **AAR-Qwen-2.5-7B** | 61.05 | 49.65 | 54.13 | 57.31 | 52.82 | 54.43 |
| w/o thinking | 53.24 | 44.51 | 51.82 | 44.91 | 43.51 | 49.23 |
| w/o analogizing | 44.12 | 39.73 | 43.51 | 42.26 | 40.12 | 42.21 |
| w/o contextualizing | 57.53 | 46.37 | 52.23 | 49.83 | 44.23 | 50.78 |
| w/o learning stage | 57.26 | 44.18 | 48.19 | 53.36 | 46.75 | 49.67 |
| w/o internalizing | 59.18 | 47.25 | 52.83 | 55.16 | 49.18 | 51.23 |
| **AAR-Llama3.1-8B** | 60.81 | 47.58 | 53.26 | 54.31 | 46.21 | 54.34 |
| w/o thinking | 53.46 | 42.25 | 46.31 | 49.13 | 43.26 | 48.13 |
| w/o analogizing | 48.39 | 38.26 | 42.17 | 42.35 | 36.18 | 40.17 |
| w/o contextualizing | 55.18 | 44.39 | 47.18 | 53.36 | 44.19 | 48.57 |
| w/o learning stage | 55.38 | 44.19 | 49.67 | 49.18 | 42.61 | 49.76 |
| w/o internalizing | 57.12 | 45.26 | 50.67 | 53.41 | 44.38 | 50.18 |

## 4.5 SCALING ANALYSIS ON NUMBER OF ANALOGICAL STEPS

To systematically investigate the impact of analogical reasoning steps on model performance, we conduct experiments on four multi-hop QA datasets (NQ, HotpotQA, MuSiQue, 2WikiQA) using Qwen-2.5-7B and Llama-3.1-8B as backbone models. We vary the number of analogical steps (1, 2, 3, 5, 10) and report F1 scores for each setting. As shown in Figure 3, increasing the number of analogical steps consistently improves model performance, especially on complex datasets such as HotpotQA and MuSiQue. When the number of steps increases from 1 to 5, the average F1 score rises by 4–5 points; however, further increasing to 10 steps yields marginal gains or even slight drops, suggesting that excessive analogical steps may introduce noise. Furthermore, Appendix A.9 provides a cost-performance analysis showing that accuracy plateaus after 5 steps, while retrieval overhead

A disease has a prevalence of 2% in a target population. A two-step screening strategy is used: Test A (rapid, inexpensive): sensitivity 0.85, specificity 0.95. Test B (confirmatory, expensive): sensitivity 0.98, specificity 0.99. Workflow: apply A to everyone; only those A-positive then take B.
Question: For a subject who is A-positive and then B-positive, what is the approximate posterior probability of disease (PPV)? Compared with the strategy "do B only once for everyone," which is better when considering both positive precision and population-scale cost?

========================================== **Turn 1** ==========================================

**< thinking >**
Cast the problem as sequential Bayesian updating: prior = population prevalence p = 0.02; observe A+, then observe B+. Use likelihood ratios (LRs) in the odds form for robust chaining.
I recall an internal pattern: the odds form of Bayes for stepwise updates—"posterior odds = prior odds × LR+"; multiple observations add in log-odds space.

**< contextualizing >**
Compute prior odds p/(1–p) = 0.02/0.98 ≈ 0.0204. For A: LR+ = sensitivity / (1–specificity) = 0.85 / (1–0.95) = 0.85/0.05 = 17.
Updated odds after A+: 0.0204 × 17 ≈ 0.347. Convert to probability: 0.347 / (1 + 0.347) ≈ 0.258 (≈ 25.8% after A+).

========================================== **Turn 2** ==========================================

**< thinking>**
Now update on B+ given A+. Compare the two-step chain with the "B-only" strategy in terms of PPV and cost.

**< analogizing >**
[1] "With serial testing, overall LR+ approximately multiplies: A's LR+ × B's LR+, which can substantially raise PPV …"
[2] "When the base rate (prevalence) is low, even a single high-performance test can yield a modest PPV; doing a cheap screen first then a confirmatory test raises positive precision …"
[3] "For cost-sensitive protocols, apply an inexpensive screen to all, then confirm positives with an expensive test—this maintains high PPV while reducing total cost …"

**< contextualizing >**
For B: LR+ = 0.98 / (1–0.99) = 98. Chained odds after B+: 0.347 × 98 ≈ 34.0.
Final posterior after A+ then B+: 34 / (1 + 34) ≈ 0.971 (≈ 97.1%). Compare with "B-only": prior odds 0.0204 × 98 ≈ 2.0 → probability 2/(1+2) = 0.667 (≈ 66.7%).
So the two-step PPV is clearly higher.
Back-of-envelope cost (per 1,000 people): Do A for all: $5 × 1,000 = $5,000. A positives ≈ 66 (true + false): with prevalence 2%, A's sensitivity/specificity give ≈ 17 true positives + 49 false positives → 66 total.
•Confirm A+ with B: $50 × 66 = $3,300. Two-step total ≈ $5,000 + $3,300 = $8,300. B-only for all: $50 × 1,000 = $50,000. Thus two-step is vastly cheaper at population scale.

========================================== **Turn 3** ==========================================

**< thinking >**
Place precision and cost into a simple decision frame, identify the mid-trajectory breakthrough, and conclude.

**< analogizing >**
[1] "With decision thresholds defined by expected utility or cost ratio, a high overall LR+ from serial testing more easily surpasses the detection threshold …"
[2] "Adding evidence in log-odds gives linear accumulation, clarifying trajectory coherence …"
[3] "Sequential strategies (screen then confirm) are standard in clinical practice and particularly advantageous under low prevalence …"

**< contextualizing >**
The mid-trajectory breakthrough occurs in Turn 2: external analogies ("multiply LR+" and "sequential is better at low base rate") upgrade the trajectory from a moderate posterior after A+ (≈ 26%) to a high posterior after B+ (≈ 97%), while also producing a dramatic cost advantage.

< Final Answer >
For a subject with A+ then B+, the posterior disease probability is ≈ 97%. Compared with "B-only" (PPV ≈ 67%), the two-step sequential strategy dominates on precision and population cost.

Figure 4: A typical case of agentic analogical reasoning trajectory.

 Overall, using 5 analogical steps achieves the best balance between performance and efficiency, and thus we adopt this setting for our main experiments.

## 4.6 CASE STUDIES

To gain deeper insights, we conduct qualitative case studies using Qwen2.5-7B, with concrete cases provided in Appendix A.4. Taking a medical screening problem as an illustrative example (Figure 4), we observe how AAR achieves progressive analogical refinement through multi-turn reasoning: (1) **Dynamic Analogy and Multi-turn Integration.** In Turn 1, when facing a two-step disease screening scenario, the model dynamically identifies the need for analogical support by generating the query "Cast the problem as sequential Bayesian updating." This exemplifies our *thinking* mechanism's ability to adaptively trigger analogical queries based on the current reasoning state, rather than relying on pre-configured templates. The model then analogizes to general statistical patterns ("prior-likelihood-posterior chains") and contextualizes this by computing the posterior probability after Test A ( 25.8%). (2) **Iterative Analogical Refinement and Knowledge Integration.** In Turn 2, the reasoning demonstrates the capacity of AAR for iterative refinement. The model transitions from broad statistical analogies to domain-specific medical screening patterns: "likelihood ratios multiply in serial testing" and "cost-sensitive screening protocols." This progression illustrates how our *analogizing* mechanism triggers increasingly relevant analogical scenarios, while the *contextualizing* phase extracts actionable insights (final PPV 97.1% and cost comparison), converting abstract analogies into concrete computational steps.

We further analyze this process through the lens of Structure-Mapping Theory (SMT) (Gentner, 1983), which posits that analogy involves mapping relational structures from a familiar *source domain* to a *target domain*. In our case, the retrieved "serial testing" protocol serves as the source, while the specific disease problem is the target. The statement "LR multiplies" represents the core *relational structure* of the source. Therefore, the *contextualizing* action effectively performs *structural alignment*: it maps this multiplicative relationship to the target's specific objects (associating Turn 1's odds with $LR_A$ and Test B with $LR_B$), thereby inferring the solution through deep structural transfer rather than surface matching.

## 5 CONCLUSION

In this paper, we proposed a novel Agentic Analogical Reasoning (AAR) paradigm, which treated the LLM as an agentic reasoner to integrate multi-turn insights along the reasoning trajectory by iteratively generating analogical queries to trigger internal or external knowledge for analogical exemplification, and selectively identifying appropriate analogies to conduct further reasoning. Specifically, we formalized an agentic analogical reasoning trajectory composed of multiple continuous analogical units. In each turn, we executed three actions as an analogical unit: (1) thinking, where analogical queries were generated adaptively based on the current reasoning state; (2) analogizing, where analogical knowledge instances were triggered from internal or external sources; and (3) contextualizing, where appropriate analogies were filtered to derive insights from analogical scenarios for the current context. To equip LLMs with AAR capability, we designed an analogical trajectory optimization algorithm. We first triggered external analogy knowledge to generate reasoning trajectories and leveraged the LLM to assign importance weights to each trajectory based on its support for the correct answer. Then, we integrated trajectory reweighting into the ELBO objective function to encourage the generation of more supportive trajectories for obtaining accurate answers. Furthermore, we devised a mixed training strategy that progressively enhanced the intrinsic analogical capabilities of LLMs by leveraging both self-generated and externally retrieved analogical trajectories. We conducted extensive experiments on seven reasoning-intensive datasets and achieved significant performance improvements over previous SOTA methods.

## 6 ETHICS STATEMENT

We follow the ICLR Code of Ethics and are committed to ensuring that our research fosters beneficial societal outcomes while carefully addressing possible risks, such as issues of privacy, bias, or unfair treatment. We maintain scientific integrity through transparent reporting, reproducibility, and honest discussion of limitations. Where relevant, we indicate the status of ethical review processes (e.g., IRB approval). We also disclose potential conflicts of interest, uphold fairness and inclusivity, and appropriately acknowledge prior contributions while respecting confidentiality and data usage agreements.

## 7 REPRODUCIBILITY STATEMENT

To enable reproducibility, we outline the following practices:

1. **Datasets.** All datasets used are publicly accessible. Detailed descriptions are provided in Section 4.1, and Appendix A.5.

2. **Baselines.** We explain the setup and hyperparameters of the baselines in Section 4.2 and Appendix A.5.

3. **Methodology.** The proposed method is described thoroughly in Section 3. Hyperparameter configurations are provided in Section A.5.2, and ablation results are included in Section 4.4 and Appendix A.5.

4. **Open Source.** We release our code and processed datasets at https://anonymous.4open.science/r/ICLR-8381.

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

# A APPENDIX

## A.1 STATEMENT ON THE USE OF LARGE LANGUAGE MODELS

In line with ICLR 2026 guidelines, we disclose the use of Large Language Models (LLMs) as writing assistants during manuscript preparation. Specifically, LLMs were employed to refine wording, improve grammar, and enhance clarity of exposition, without influencing the scientific content.

## A.2 PROMPT INSTRUCTION FOR AAR

```
You are a reasoning assistant that uses multi-turn analogical reasoning
to answer the user's question accurately.
Given a question, iteratively perform the following three actions for up
to 5 turns:

(1) <thinking>: Formulate an analogy query that pinpoints which aspect of
 the current reasoning needs analogical support.
(2) <analogizing>: Retrieve 5 relevant analogical scenarios from external
 knowledge. This will trigger the retrieval system and return relevant
documents. If you can generate analogical scenarios from internal
knowledge instead, skip this step and directly describe the analogical
case (e.g., "I recall a similar problem involving..."). In both cases,
prefer deep relational similarity over surface similarity.
(3) <contextualizing>: Distill the transferable insight from the
analogical scenarios (whether from external retrieval or internal
generation) for the current problem and state how it could be used in the
 current reasoning process.

Interleave <thinking>, <analogizing>, and <contextualizing> repeatedly
until you can produce the final answer.
Conclude your output with:
Final Answer: <the final answer>

Formatting rules:
- When using external retrieval: Each action must start on its own line
with the exact token: "<thinking>", "<analogizing>", or "<contextualizing
>".
- When using internal analogies: Use "<thinking>" and "<contextualizing>"
 tokens, but skip the "<analogizing>" token and directly output your
internal analogical case.
```

```
- Keep each action concise and atomic (no markdown, no extra commentary).
- Be explicit in <contextualizing> about the insight and the concrete
update to the intermediate state.
- You may refine or replace previously triggered analogies in later turns
 if they become unhelpful.

Below is the task for you to complete:

<USER QUERY> {task}
Your Output:
```

## A.3 PROMPT TEMPLATE FOR INTERNALIZATION

```
You are tasked with converting external analogical reasoning steps into
internal analogical reasoning patterns. Given an analogical reasoning
trajectory that contains external retrieval steps (marked with <
analogizing> tokens), transform these external analogies into internal,
self-generated analogical patterns.

**Conversion Rules:**
1. Replace external retrieval statements with internal recall patterns
starting with "I recall a similar problem..." or "This reminds me of..."
2. Maintain the core analogical content and relational structure
3. Keep the same reasoning flow and insights
4. Preserve all <thinking> and <contextualizing> steps unchanged

**Examples:**
- External: "<analogizing>: An analogous problem is the Tower of Hanoi
puzzle..."
- Internal: "I recall a similar problem, the Tower of Hanoi puzzle..."

- External: "<analogizing>: This problem is similar to the traveling
salesman problem..."
- Internal: "This reminds me of the traveling salesman problem..."

**Instructions:**
Given the following analogical reasoning trajectory with external
retrieval steps, convert it to an internal analogical reasoning pattern
while preserving all the reasoning content and insights.

**Input Trajectory:**
{trajectory}

**Output Requirements:**
- Remove some <analogizing> tokens
- Convert external analogical statements to internal recall patterns
- Keep all other reasoning steps (thinking, contextualizing) exactly the
same
- Maintain the same logical flow and conclusions

**Converted Trajectory:**
```

## A.4 CASE STUDIES

In the following, we present qualitative case studies to illustrate how our iterative *think–analogize–contextualize* paradigm unfolds in practice. Each case is organized as follows:

- **Task**: We first state the problem to be solved, providing a concise description of the input and the expected output.
- **Analogize**: For each reasoning turn, we retrieve three related analogical cases from the corpus. Only the essential sentences are retained, and minor details are shortened with ellipses "…". These

analogies serve to trigger relevant knowledge and connect the current problem to prior structured reasoning experiences.

- **Contextualize**: After each analogical step, t he extracts key insights and instantiates them as constraints or intermediate conclusions. This ensures that analogies are not only recalled but also adapted into the evolving reasoning trajectory.

- **Role of Analogy in Contextual Reasoning**: By embedding analogical insights into the trajectory, the LLM maintains coherence across turns, dynamically updating intermediate states and reducing premature or inconsistent choices.

- **Result**: We highlight the final aggregated answer, showing how intermediate insights converge.

- **Original I/O**: Finally, we provide the raw multi-turn input–output transcript (*THINK / Analogize / Contextualize / Final Answer*), with only minor ellipses inside the analogical quotes. Boldface marks the cross-referential keywords that align with our introduction (e.g., **endo**, **secondary orbital interactions**, **CRT merge**), indicating the extracted key insights.

**Case 1**

- Task: Determine the *major stereoisomer* after a three-step sequence (COT + maleic anhydride; MeOH/$H_2SO_4$; cyclopentadiene).

- Triggering analogical knowledge instances:

  [1] "... under kinetic control the endo adduct dominates due to secondary orbital interactions ... forming a bridged adduct ..."

  [2] "... acid-catalyzed methanolysis gives the dimethyl diester with cis retained ..."

  [3] "... on preorganized bicyclic maleate/fumarate scaffolds, endo approach relieves bridge strain ..."

- Contextualizing (extracting key insights $\kappa$ and instantiating constraints): $\kappa$ = {endo, secondary orbital interactions, cis-retaining methanolysis, minimal bridge strain};
  C1 "Step 1 choose endo"; C2 "anhydride $\rightarrow$ dimethyl diester with cis retained"; C3 "Step 3 endo to reduce bridge strain".

- Mid-trajectory breakthrough: at Turn 3 (C1+C2 define the framework; C3 unifies orientation and strain).

- Outcome: the fused major isomer consistent with two endo Diels–Alder steps.

```
INPUT QUERY:
Given a 3-step sequence: (1) COT + maleic anhydride (1:1, heat) ->
product 1;
(2) product 1 + MeOH, cat. H2SO4 (heat) -> product 2;
(3) product 2 + cyclopentadiene (heat) -> product 3.
Question: Which is the major stereoisomer? (Gold answer: [fill])

============ Turn 1 ============
< THINK >
Decide whether step (1) is a Diels--Alder (DA), the endo/exo preference,
and whether a bridged framework forms.
<Analogize>
[1] Cyclopentadiene + maleic anhydride under kinetic conditions gives
predominantly endo, driven by secondary orbital
interactions, forming a bridged norbornene-type adduct ...
[2] Furan + maleic anhydride is a classic DA pair; maleic anhydride is a
strong dienophile, with bias toward endo even at lower temperature ...
[3] Anthracene (9,10-diene) + maleic anhydride reacts at 9,10; the
initial adduct frequently favors endo, influenced by secondary orbital
interactions ...
<Contextualize>
Treat COT as the diene toward maleic anhydride; predict an endo-set
bridged bicyclic adduct (product 1).

============ Turn 2 ============
< THINK >
Identify the transformation and configurational fate in step (2) under
MeOH / cat. H2SO4.
<Analogize>
```

```
[1] Norbornene -5,6-endo-dicarboxylic anhydride undergoes acid-catalyzed
methanolysis to the corresponding dimethyl
diester, with adjacent stereocenters cis
retained; no bridgehead epimerization under standard conditions ...
[2] Phthalic anhydride converts cleanly to the dimethyl
diester in MeOH/H2SO4, illustrating the general acidic alcoholysis from
anhydride to diester ...
[3] cis-1,2-Cyclohexanedicarboxylic anhydride gives the dimethyl
diester while retaining cis ...
<Contextualize>
Convert the anhydride in product 1 via methanolysis to the dimethyl
diester (product 2), preserving the cis relationship set in step (1).

============ Turn 3 ============
< THINK >
Assess step (3): cyclopentadiene addition to product 2 (electron-poor
alkene), orientation and strain effects.
<Analogize>
[1] Cyclopentadiene + dimethyl maleate shows pronounced endo selectivity
under kinetic control; secondary orbital
interactions stabilize the endo TS of electron-poor alkenes ...
[2] Cyclopentadiene + maleimide strongly favors endo on heating, giving a
norbornene-type adduct ...
[3] On preorganized bicyclic maleate/fumarate esters, endo approach
alleviates bridge
strain, often yielding a single dominant orientation ...
<Contextualize>
On the electron-poor
alkene of product 2, an endo trajectory both maximizes secondary orbital
interactions and minimizes bridge strain, fixing new stereocenters and
yielding the dominant stereoisomer (product 3).

============ Turn 4 (answer aggregation) ============
Final Answer:
The fused endo-consistent major stereoisomer (fill in the exact label/
depiction).
```

**Analysis**: This case exemplifies **multi-turn integration** through coherent analogy-driven trajectories. The model seamlessly transitions from Diels-Alder analogies in Turn 1 to methanolysis patterns in Turn 2, and finally to strain minimization in Turn 3. Each turn builds upon previous insights while introducing new analogical perspectives, demonstrating how AAR maintains reasoning coherence across multiple *think–analogize–contextualize* cycles without losing track of the overall synthetic pathway.

**Case 2**

- Task: Compute the *last three digits* of $7^{999}$.
- Triggering analogical knowledge instances:

    [1] "Modulus decomposition: $1000 = 8 \times 125$, and coprime — solve on each modulus ..."

    [2] "On mod 125, use Euler/cycle for exponent reduction ... '$7^{\phi(125)} \equiv 1$' ..."

    [3] "CRT merging: use modular inverses to solve $x \equiv a \pmod 8$, $x \equiv b \pmod{125}$ ..."

- Contextualizing (extracting key insights $\kappa$ and instantiating constraints): $\kappa = \{$modulus decomposition, exponent reduction, CRT merge/modular inverses$\}$;
  C1 "run two congruences in parallel"; C2 "do exponent reduction on mod 125 first"; C3 "use CRT/modular inverses to obtain the unique solution".
- Mid-trajectory breakthrough: Turn 3 closes the loop from decomposition to merging.
- Outcome: last three digits = [gold].

```
INPUT QUERY:
Compute the last three digits of 7^{999} (Gold answer: [fill]).

============ Turn 1 ============
< THINK >
```

```
Decompose 1000 into coprime factors and plan to compute on each modulus
separately.
<Analogize>
[1] "Modulus decomposition: 1000 = 8 \times 125, and coprime; compute modulo 8
 and modulo 125 ..."
[2] "On mod 125, use Euler's function / cycle for exponent reduction: '\
ldots 7^{\phi(125)} \equiv 1 \ldots' combining period and exponent
reduction ..."
[3] "On mod 8, 7 \equiv -1, so odd powers remain 7 (\equiv -1), then use
CRT
merging together with the high-mod result ..."
<Contextualize>
Set subgoals: a = 7^{999} (mod 8); b = 7^{999} (mod 125).

============ Turn 2 ============
< THINK >
Carry out exponent reduction on mod 125; simplify directly on mod 8.
<Analogize>
[1] "Since 125 = 5^3, \phi(125) = 100; thus 7^{999} = 7^{100\times 9+99}
\equiv 7^{99} (mod 125) ..."
[2] "Use binary exponentiation or a known power table, ensuring
intermediate reductions under mod
125 ..."
[3] "On mod 8, 7^{odd} \equiv 7 ..."
<Contextualize>
Obtain a = [fill] (mod 8), b = [fill] (mod 125).

============ Turn 3 ============
< THINK >
Apply the Chinese Remainder Theorem to combine congruences and get the
unique solution modulo 1000.
<Analogize>
[1] "CRT merge: solve x \equiv a \pmod{8}, x \equiv b \pmod{125} ..."
[2] "Via modular inverses: set M1 = 125, M2 = 8; compute M1^{-1} \pmod{8}
and M2^{-1} \pmod{125} ..."
[3] "x = a\cdot M1\cdot inv1 + b\cdot M2\cdot inv2 \pmod{1000}, then
simplify to the unique solution ..."
<Contextualize>
Substitute a, b and the inverses to get x (mod 1000) = last three digits
= [fill].

============ Turn 4 (answer aggregation) ============
Final Answer:
Last three digits = [fill] (obtained via the chain "modulus
decomposition \rightarrow exponent reduction \rightarrow CRT merge/modular
inverses").
```

**Analysis**: This case demonstrates the **explicit analogical chain** from analogy to actionable constraints. In Turn 1, the "modulus decomposition" analogy directly translates to the executable constraint of computing separately modulo 8 and 125. The model explicitly articulates how the analogical insight (coprime factor decomposition) becomes a concrete computational strategy, illustrating the transparent "analogy → insight → constraint" pathway that enhances both reliability and interpretability.

## A.5 MORE EXPERIMENT DETAILS

### A.5.1 BASELINE SETTINGS

We implement the above baselines following their official code. For fair comparison, we use the same retrieval corpus as our method for all retrieval-based baselines. Full model descriptions are included in Appendix A.5. Following the most commonly used recipe for baselines, we set the size of retrieval documents to 10 for all retrieval-augmented baselines and 5 for multi-turn retrieval methods (as they can retrieve multiple times), as well as for our method. For baselines including RankRAG (Yu et al., 2024) without publicly available code, we report results from their original papers but only for reference (marked as "-" for metrics that were not reported in the original paper).

### A.5.2 IMPLEMENTATION DETAILS

We implement AAR on Qwen-2.5-7B-Instruct and Llama-3.1-8B. Following prior work (Xu et al., 2024), we build by combining the December 20, 2018 Wikipedia passage dump with reasoning trajectories from ReasonFlux (Yang et al., 2025) and OpenThoughts (Guha et al., 2025). We adopt ColBERTv2.0 (Santhanam et al., 2021) for document retrieval. To equip the model with the basic ability to follow the "thinking–analogizing–contextualizing" routine, we first synthesize 1,000 pseudo-examples and perform a small-scale fine-tuning under a standard cold-start setup, following a cold-start setup as in prior work (Chen et al., 2025b; Song et al., 2025). For subsequent trajectory generation, we set the number of iterations $N=5$ and report model performance at each iteration. The preliminary phase iteration $N_e$ is set as 2. Our training data incorporates the official training sets provided by Natural Questions (NQ), HotpotQA and ReasonFlux. For multi-hop QA tasks, we merged the training sets of NQ and HotpotQA to form a unified dataset for training, and evaluation was conducted on the test sets of four datasets to assess both in-domain and out-of-domain performance.

### A.6 MORE RELATED WORKS

**Advanced Reasoning in LLMs** Large Language Models (LLMs) have demonstrated remarkable capabilities in reasoning, solving complex problems by exhibiting traits such as reflection, backtracking, and multi-path thinking (Wang et al., 2022a; Chen et al., 2024; Kumar et al., 2024; Wu et al., 2024; Ma et al., 2025). This ability is attributed to higher-order reasoning capabilities that approximate human cognition, marking a shift from merely answering questions to demonstrating more complex thought patterns (Chen et al., 2025a). Along this line of research, some work has sought to further enhance cognitive reasoning abilities. For instance, training in inductive-deductive skills has been used to bolster the model's capacity to solve complex mathematical and scientific problems (Hu et al., 2025). However, enabling models to perform analogical reasoning—where humans draw from relevant past experiences to tackle new problems —remains a challenge Yasunaga et al. (2023b).

**Analogical Reasoning** Recent efforts to endow LLMs with analogical reasoning capabilities have drawn inspiration from classical cognitive theories such as structure-mapping (Gentner, 1983; Hofstadter & Sander, 2013). Early computational studies mainly tested lexical or relational analogies (Chen et al., 2022; Czinczoll et al., 2022), while recent advances have extended analogical reasoning to richer targets such as systems Yuan et al. (2023), processes Bhavya et al. (2022), paragraphs Webb et al. (2022), and stories Jiayang et al. (2023). Methods like Analogical Prompting (Yasunaga et al., 2023b), Meta-Ability Alignment (Hu et al., 2025), Thought Propagation (Yu et al., 2023), and Thought Retriever (Feng et al., 2024) further embed analogy generation into reasoning chains, prompting models to generate or retrieve analogous problems to assist in solving the current one. However, these approaches often remain at the surface level, suffering from noise accumulation or dependence on external knowledge bases, and thus still fall short of enabling deep, human-like cross-contextual reasoning. Developing a more general and robust framework for analogical reasoning in complex tasks, therefore, remains an open challenge.

### A.7 IMPLEMENTATION DETAILS OF GRADIENT OPTIMIZATION

The gradient in Equation 5, $\nabla_\theta E_{\boldsymbol{z} \sim p(\boldsymbol{z}|x;\theta^t)} [w(\boldsymbol{z}) (\log p(\boldsymbol{z}|x;\theta) + \log p(\boldsymbol{y}|x,\boldsymbol{z};\theta))]$, is approximated using a sample-based approach, which is standard for optimizing expectations in variational inference or reinforcement learning. In practice, for each training instance (x, y), we perform the following steps:

1. **Sampling Trajectories:** We first sample one or more reasoning trajectories, $\boldsymbol{z} \sim p(\boldsymbol{z}|x;\theta^t)$, using the current model parameters $\theta^t$ (i.e., a forward pass in inference mode).

2. **Calculating Importance Weights:** For each sampled trajectory $\boldsymbol{z}$, we compute its importance weight $w(\boldsymbol{z})$. As defined in Section 3.2, this weight is proportional to $p(y|x,\boldsymbol{z};\theta^t)$, which is the log-probability of generating the ground-truth answer $y$ given the input $x$ and the sampled trajectory $\boldsymbol{z}$. This weight is treated as a fixed scalar for the subsequent loss calculation.

3. Computing the Gradient via Weighted Loss:
   The core of the implementation lies in interpreting the two log-probability terms, $\log p(\boldsymbol{z}|x;\theta)$ and $\log p(y|x,\boldsymbol{z};\theta)$, as a standard auto-regressive language modeling objective.

- The trajectory $z$ is a sequence of tokens. As per Equation 1, its log-probability, $\log p(z|x; \theta)$, is the sum of the log-probabilities of generating each token in the sequence.

- Similarly, $\log p(y|x, z; \theta)$ is the log-probability of generating the answer tokens $y$ after the model has been conditioned on $x$ and $z$.

- In practice, we can combine these two terms by simply concatenating the trajectory and the answer into a single target sequence: $[z, y]$. The entire expression $\log p(z|x; \theta) + \log p(y|x, z; \theta)$ then becomes the standard cross-entropy loss for generating the sequence.

The gradient of this loss is then computed with respect to the model parameters $\theta$ and used for the SGD update.

### A.8 COMPUTATIONAL EFFICIENCY ANALYSIS

In this section, we provide a detailed analysis of the computational budget and inference latency. The memory usage consists of a static part for the model parameters, which is identical across all methods due to the unmodified backbone, and a dynamic portion that is proportional to the length of generated output tokens.

To provide a fair comparison, we evaluated AAR against the strong trainable baselines from our main experiments: the retrieval-augmented method Search-R1 and the analogical reasoning method Meta-Ability Alignment. All methods were implemented using the same Qwen-2.5-7B backbone. Table 4 summarizes the training and inference costs on the HotpotQA dataset.

Table 4: Comparison of computational budget, inference latency (measured by token usage and retrieval calls), and performance on HotpotQA using Qwen-2.5-7B backbone.

| Method | Training Hours | Avg. Input Tokens | Avg. Output Tokens | Avg. Total Tokens | Retrieval Calls | Accuracy (%) |
|---|---|---|---|---|---|---|
| **AAR (Ours)** | **8.2** | **1870.18** | 330.21 | **2200.39** | **2.55** | **54.13** |
| Meta-Ability Alignment | 11.8 | 2143.68 | 301.25 | 2444.93 | 3.04 | 44.26 |
| Search-R1 | 16.4 | 2355.12 | 314.26 | 2669.38 | 3.61 | 52.46 |

Compared to the direct reasoning capability of the vanilla backbone (Acc. 18.31%), these fine-tuned methods demonstrate clear performance gains. Among them, AAR achieves the **highest accuracy (54.13%)** while requiring significantly **fewer training hours (8.2)** compared to both Meta-Ability Alignment (11.8) and Search-R1 (16.4). For inference, AAR demonstrates **superior efficiency by consuming fewer tokens and requiring fewer retrieval calls**. The "Retrieval Calls" metric indicates the average number of times the model invokes the external retrieval system per query. AAR's lower value (2.55) suggests its "thinking-analogizing-contextualizing" mechanism more effectively determines when to seek external knowledge, avoiding redundant searches. This leads to fewer retrieval calls and lower total token consumption, resulting in lower inference latency compared to the baselines.

### A.9 TRADE-OFF ANALYSIS ON ANALOGICAL STEPS

In Section 4.5, we presented the performance scaling with respect to the number of analogical steps. Here, we provide a detailed analysis of the trade-off between accuracy gains and computational cost. As shown in Table 5, increasing the maximum steps from 1 to 5 yields a substantial F1 score improvement (from 51.00 to 61.05) with a moderate increase in computational effort (average retrieval calls rising to 2.55).

However, extending the steps further from 5 to 10 drastically raises the average retrieval calls to 6.55, significantly increasing inference latency and computational overhead. Despite this added cost, performance slightly degrades (dropping to 60.79), likely due to noise accumulation from excessive retrieval. This analysis confirms that setting the limit at 5 steps strikes the optimal balance between reasoning performance and computational feasibility.

Table 5: Trade-off analysis between accuracy (F1 Score) and computational cost (Avg. Retrieval Calls) on HotpotQA.

| Maximum Steps | F1 Score (%) | Avg. Retrieval Calls |
|---|---|---|
| 1 | 51.00 | 1.00 |
| 5 | 61.05 | 2.55 |
| 8 | 60.90 | 4.95 |
| 10 | 60.79 | 6.55 |

Table 6: Performance comparison of AAR equipped with different retrieval backbones on HotpotQA.

| Retriever | F1 Score | EM Score |
|---|---|---|
| ColBERTv2 | 61.05 | 49.65 |
| E5 | 60.89 | 49.41 |
| DPR | 60.59 | 48.91 |
| Contriever | 46.52 | 35.62 |
| BM25 | 45.18 | 34.89 |

## A.10 IMPACT OF DIFFERENT RETRIEVAL MODELS

AAR is designed as a modular reasoning framework, making its core "thinking-analogizing-contextualizing" mechanism largely agnostic to the specific retriever used. To empirically validate this robustness, we conducted additional experiments on HotpotQA by replacing the default ColBERTv2 with other representative retrievers: BM25 (sparse), Contriever (unsupervised dense), DPR, and E5 (supervised dense). The number of retrieved documents remains fixed at 5.

As shown in Table 6, AAR demonstrates consistent high performance with strong supervised dense retrievers. Specifically, E5 and DPR achieve F1 scores (60.89 and 60.59) comparable to ColBERTv2, indicating that our framework effectively leverages high-quality semantic retrieval regardless of the specific architecture. While performance drops with weaker retrievers like BM25 and Contriever, this is expected as the reasoning quality depends on the relevance of the retrieved analogical context. Overall, the results confirm that AAR is compatible with various retrieval pipelines.

