# OpenReview forum: "Agentic Analogical Reasoning for Large Language Models"
_ICLR.cc/2026/Conference — Submitted to ICLR 2026_

### Official Review · Reviewer_m7U5 · 2025-10-29

**Soundness:** 4
**Presentation:** 3
**Contribution:** 3
**Rating:** 6
**Confidence:** 4

**Summary:**

This paper proposes a novel Agentic Analogical Reasoning (AAR) paradigm to enhance LLMs’ analogical reasoning capability. The authors define key processes to complete analogical reasoning. The model generates a sequence of intermediate reasoning steps before producing the final answer.The idea of formulating this process as an ELBO optimization problem is elegant and theoretically sound.

**Strengths:**

1. Equations (1)–(6) are clearly presented and easy to follow.
2. The formalization of analogical reasoning as ELBO optimization is both creative and technically coherent.
3. The paper includes extensive experiments on reasoning-intensive datasets, showing consistent gains over prior SOTA methods.

**Weaknesses:**

1. **Main Issue**. It is unclear how Equation (7) is concretely implemented in the model. Specifically, since it involves $\nabla_\theta (logp(z∣x;θ)+logp(y∣x,z;θ))$, the paper should describe in detail (or provide pseudocode) how this gradient-based optimization is realized in practice.
2. The paper lacks clear information about training data. It is important to clarify what training corpus was used for model learning.
3. According to the supplementary material,$ a=A(q)$ where $A$ denotes a retriever. Hence $a$ is not generated by the model parameters $\theta $. However, in Equation (7),  $a$  is required to be generated by $\theta $. How is this inconsistency handled during training?
4. Figures 1 and 2 appear to be unreferenced in main text.

**Questions:**

See weakness. I will **re-evaluate my recommendation after the authors provide clarifications and additional important details**.

---

> ### Author Response · Authors · 2025-11-21
> **Response to Reviewer m7U5 (1/2)**
>
> Thanks for your constructive comments! We address your concerns or questions as follows.
>
> > ***W1 (Main Issue): It is unclear how Equation (7) is concretely implemented in the model. Specifically, since it involves $\nabla_\theta(\log p(z \mid x; \theta) + \log p(y \mid x, z; \theta))$ , the paper should describe in detail (or provide pseudocode) how this gradient-based optimization is realized in practice.***
>
> **Response:** We appreciate the comment and clarify the details of Equation (7). The gradient in Equation (7), $\nabla_{\theta} E_{\boldsymbol{z}\sim p(\boldsymbol{z}|x;\theta^{t})}\left[w(\boldsymbol{z})\left(\log p(\boldsymbol{z}|x;\theta) + \log p(\boldsymbol{y}|x,\boldsymbol{z};\theta)\right)\right]$, is approximated using a sample-based approach, which is standard for optimizing expectations in variational inference or reinforcement learning. In practice, for each training instance (x, y), we perform the following steps:
>
> 1. **Sampling Trajectories:** We first sample one or more reasoning trajectories, $\boldsymbol{z} \sim p(\boldsymbol{z}|x; \theta^t)$, using the current model parameters $\theta^t$ (i.e., a forward pass in inference mode).
>
> 2. **Calculating Importance Weights:** For each sampled trajectory $\boldsymbol{z}$, we compute its importance weight $w(\boldsymbol{z})$. As defined in Section 3.2, this weight is proportional to $p(y|x, \boldsymbol{z}; \theta^t)$, which is the log-probability of generating the ground-truth answer y given the input x and the sampled trajectory $\boldsymbol{z}$. This weight is treated as a fixed scalar for the subsequent loss calculation.
>
> 3. Computing the Gradient via Weighted Loss:
>
>     The core of the implementation lies in interpreting the two log-probability terms, $\log p(\boldsymbol{z}|x;\theta)$ and $\log p(y|x,\boldsymbol{z};\theta)$, as a standard auto-regressive language modeling objective.
>
>    - The trajectory $\boldsymbol{z}$ is a sequence of tokens. As per Equation (1), its log-probability, $\log p(\boldsymbol{z}|x;\theta)$, is the sum of the log-probabilities of generating each token in the sequence.
>    - Similarly, $\log p(y|x,\boldsymbol{z};\theta)$ is the log-probability of generating the answer tokens y after the model has been conditioned on x and z.
>    - In practice, we can combine these two terms by simply concatenating the trajectory and the answer into a single target sequence: [z, y]. The entire expression $\log p(\boldsymbol{z}|x;\theta) + \log p(y|x,\boldsymbol{z};\theta)$ then becomes the standard cross-entropy loss for generating the sequence.
>
> The gradient of this loss is then computed with respect to the model parameters $\theta$ and used for the SGD update.
>
> > ***W2: The paper lacks clear information about training data. It is important to clarify what training corpus was used for model learning.***
>
> **Response:** Sorry for the confusion. Due to space limitations, the full details of the knowledge base were included in Appendix A.5.2 (Implementation Details) in lines 1000-1010. Specifically, our training data incorporates the official training sets provided by Natural Questions (NQ), 2WikiMultiHopQA and ReasonFlux. We use a hybrid corpus by combining the Wikipedia passage dump from December 20, 2018, and reasoning trajectories from the ReasonFlux[1] and OpenThoughts[2] datasets as the retrieval corpus. Wikipedia is used for General multi-hop reasoning tasks, whereas ReasonFlux and OpenThoughts are used for Scientific reasoning tasks.
>
> We will further clarify these dataset details in the revised paper to ensure no ambiguity.
>
> [1]Yang, Ling, et al. "Reasonflux: Hierarchical llm reasoning via scaling thought templates." *arXiv preprint arXiv:2502.06772* (2025).
>
> [2]Guha, Etash, et al. "OpenThoughts: Data Recipes for Reasoning Models." *arXiv preprint arXiv:2506.04178* (2025).

---

> ### Author Response · Authors · 2025-11-21
> **Response to Reviewer m7U5 (2/2)**
>
> > ***W3: According to the supplementary material, $a_i = \mathcal{A}(q_i)$ where $\mathcal{A}$ denotes a retriever. Hence $a_i$ is not generated by the model parameters $\theta$. However, in Equation (7), $p(z | x; \theta)$ is required to be generated by $\theta$. How is this inconsistency handled during training?***
>
> **Response:** We appreciate this thoughtful review. There is a distinction between parts of the trajectory generated by the model's parameters $\theta$ and those provided by the external retriever. Our framework handles this by treating the retriever as a fixed, non-differentiable component of the environment, separate from the language model we are optimizing.
>
> The process is divided into two distinct phases for each training step: (i) **Trajectory Generation:** During this phase, the model autoregressively generates the trajectory $\boldsymbol{z}$. When it emits the special token <analogizing>, we pause generation, execute a call to the external retriever $\mathcal{A}(\cdot)$ with the generated query $q_i$, and then insert the retrieved text $a_i$ into the generation context. The model then continues generating the contextualizing part based on this new, augmented context. (ii) **Gradient Calculation:** During the optimization step, the gradient is only computed for the parts of the trajectory that were actually generated by the model's parameters $\theta$. Specifically, the gradient $\nabla_\theta \log p(\boldsymbol{z}|x;\theta)$ is computed over the tokens of the thinking step ($q_i$) and the contextualizing step ($\kappa_i$). The retrieved analogy text $a_i$ is treated as a fixed, given part of the input that conditions the generation of the contextualizing step ($\kappa_i$). Therefore, no gradient is computed for the text of $a_i$ itself, as it is not part of the model's generative process.
>
> This approach is analogous to how tool-augmented LLMs are trained: the model learns when to call a tool and how to use its output, but the tool itself is not part of the model's differentiable computation graph. Our retriever's parameters are independent and are not updated during this process.
>
> > ***W4: Figures 1 and 2 appear to be unreferenced in main text.***
>
> **Response:** We appreciate your suggestion! We will add the reference in the revision.

---

> ### Comment · Reviewer_m7U5 · 2025-11-24
> **Response**
>
> Upon re-evaluating the manuscript, I found that this work exhibits a very high degree of overlap with a previously published paper in NeurIPS 2026. This raises serious concerns. Specifically, the ELBO proposed by the authors appears to be almost entirely derived from [1], including the experimental design and baseline comparisons, which are strikingly similar. Superficially, it seems that Think-Search has merely been replaced with Analogical Reasoning, yet the original work is not cited anywhere in the manuscript.
>
> Given this high similarity, the authors should provide a clear and reasonable explanation regarding the novelty of this work and its relation to [1].
>
> [1] Shi Z, Yan L, Yin D, et al. Iterative self-incentivization empowers large language models as agentic searchers[J]. arXiv preprint arXiv:2505.20128, 2025.

---

> ### Author Response · Authors · 2025-11-25
> **Response to Reviewer m7U5 (1/3)**
>
> We sincerely thank you for your attention to detail and for bringing ExSearch [1] to our attention. **However, we strongly disagree with the characterization that our work is a "superficial replacement."** First, our work focuses on **improving the reasoning capabilities of LLMs** rather than enhancing RAG; therefore, our task is **fundamentally different** from ExSearch. In addition, both the technical approach and experiments proposed in our paper are also different, as outlined below:
>
> - **A New Framework for Agentic Analogy Reasoning**: Our approach draws on the principle of analogy to enhance the reasoning capabilities of LLMs. Inspired by **Structure-Mapping Theory**, we model reasoning as an **active analogy process and propose a standardized analogical unit** (thinking-analogizing-contextualizing) to optimize the reasoning trajectory. Specifically, the thinking action performs adaptive analogy query generation, and the analogizing action triggers **internal or external analogy instances for analogical exemplification**. Crucially, the contextualizing action transcends passive information extraction by selectively identifying appropriate analogies to conduct further reasoning. This step implements a structure-mapping mechanism: it **extracts insights from the source analogy instance and migrates them to the target context**. In contrast, ExSearch functions as a searcher relying **exclusively on external knowledge for factual retrieval**; its "recording" action **only records text evidence** to answer queries instead of reasoning based on the context, which fundamentally differs from our mechanism of extracting reasoning patterns from analogous scenarios.
> - **An Effective Analogical Trajectory Optimization Algorithm**: Given that directly applying traditional training algorithms to analogical trajectory optimization is challenging, we leverage the variational inference framework [21] to guide the optimization of analogical trajectories. **It is worth emphasizing that this optimization paradigm has been extensively employed in both trajectory optimization [18,19] and reinforcement learning [17,20]**. Below, we present the detailed derivation specific to our method.
> > **Theoretical Formulation.** Adopting a *variational inference* framework [21], the basic procedure is as follows: we treat the analogical reasoning trajectory $z$ as a *latent variable*. Since directly inferring the target posterior distribution $p(z|x, y)$ is intractable, the model is trained by *maximizing the ELBO* (Evidence Lower Bound), which is derived via *Jensen's inequality*. During optimization, we utilize *Importance Sampling* [12], using the prior distribution $p(z|x)$ as the sampling distribution to approximate the target posterior distribution $p(z|x, y)$. This is consistent with the PPO [17] in reinforcement learning, where samples from an old policy (analogous to the prior) are used to optimize the new policy. From a statistical perspective, this corresponds to the Generalized Expectation-Maximization (GEM) algorithm.
>
> - **A Novel Mechanism for Intrinsic Capability Enhancement**: For **progressively strengthens the model's intrinsic analogical capabilities**, we design a novel mixed training strategy. This is achieved by utilizing both self-generated and externally retrieved analogical trajectories. Our internalization process introduces a specific optimization objective focused on internalizing analogical reasoning, a feature that effectively distinguishes our work from prior studies [1,10,11].
> - Extensive Validation on Reasoning Tasks: We validate the effectiveness of our method on both knowledge-intensive and scientific reasoning tasks. Notably, **we find that our method performs particularly well on more challenging scientific reasoning tasks (GPQA, AIME2024, and AMC2023)**, achieving an average absolute improvement of 12.6% over the state-of-the-art analogical reasoning baseline, as shown in Table 2 of the manuscript.

---

> ### Author Response · Authors · 2025-12-03
> **Response to Reviewer m7U5 (2/3)**
>
> Moreover, we would like to emphasize the technical novelty of our work. We propose the Agentic Analogical Reasoning (AAR) paradigm, which is characterized by:
>
> *   It treats the LLM as an agentic reasoner capable of **integrating multi-turn insights throughout the reasoning trajectory**.
> *   Instead of relying on static prompts, this paradigm empowers the model to **dynamically generate** analogy queries that **trigger internal or external instances for analogical exemplification**, while **selectively identifying appropriate analogies** to guide subsequent reasoning.
> *   Specifically, via analogical trajectory optimization and internalization, we enable LLMs to **inherently develop this active analogical reasoning capability**.
>
> Finally, we would like to restate the motivation of our work. In February 2025, we have observed that while reasoning models like DeepSeek-R1 have significantly improved reasoning capabilities, they still struggle to resolve complex reasoning tasks (e.g. complex scientific problems) and face issues like factual hallucination and constrained reasoning context.
> Prior evaluations on conceptual analogy [2,3] and system-level analogy [4-7] demonstrate that language models possess a certain degree of analogical capability.
> Along this line, methods like Analogical Prompting [8,9] have leveraged this capability to enhance LLM reasoning by self-generating relevant examples as context before solving problems. However, prompting LLMs to directly generate answers through analogical reasoning still faces limitations, as existing approaches typically rely on shallow single-turn reasoning, making it easy for generating unreliable instances from internal knowledge, and lack mechanisms to identify appropriate analogies, which restricts their effectiveness in complex reasoning tasks. This motivates us to propose the Agentic Analogical Reasoning (AAR) paradigm, empowering LLMs to integrate multi-turn analogical insights into their reasoning process.
>
> We once again thank the reviewer for the comments. We have now revised the manuscript to more clearly present our motivation and methodology, supplemented additional experiments and analyses, and incorporated a detailed discussion of [1] to better articulate the contributions of our paper.

---

> ### Author Response · Authors · 2025-12-03
> **Response to Reviewer m7U5 (3/3)**
>
> References:
>
> [1] Shi Z, Yan L, Yin D, et al. Iterative self-incentivization empowers large language models as agentic searchers[J]. arXiv preprint arXiv:2505.20128, 2025.
>
> [2] Chen, Jiangjie, et al. "E-KAR: A benchmark for rationalizing natural language analogical reasoning." arXiv preprint arXiv:2203.08480 (2022).
>
> [3] Czinczoll, Tamara, et al. "Scientific and creative analogies in pretrained language models." arXiv preprint arXiv:2211.15268 (2022).
>
> [4] Yuan, Siyu, et al. "Beneath surface similarity: Large language models make reasonable scientific analogies after structure abduction." arXiv preprint arXiv:2305.12660 (2023).
>
> [5] Bhavya, Bhavya, Jinjun Xiong, and ChengXiang Zhai. "Analogy generation by prompting large language models: A case study of instructgpt." arXiv preprint arXiv:2210.04186 (2022).
>
> [6] Webb, Taylor, Keith J. Holyoak, and Hongjing Lu. "Emergent analogical reasoning in large language models." Nature Human Behaviour 7.9 (2023): 1526-1541.
>
> [7] Jiayang, Cheng, et al. "StoryAnalogy: Deriving story-level analogies from large language models to unlock analogical understanding." Proceedings of the 2023 Conference on Empirical Methods in Natural Language Processing. 2023.
>
> [8] Yasunaga, Michihiro, et al. "Large language models as analogical reasoners." arXiv preprint arXiv:2310.01714 (2023).
>
> [9] Qin, Chengwei, et al. "Relevant or Random: Can LLMs Truly Perform Analogical Reasoning?." Findings of the Association for Computational Linguistics: ACL 2025. 2025.
>
> [10] Li, Xiaoxi, et al. "Search-o1: Agentic search-enhanced large reasoning models." arXiv preprint arXiv:2501.05366 (2025).
>
> [11] Jin, Bowen, et al. "Search-r1: Training llms to reason and leverage search engines with reinforcement learning." arXiv preprint arXiv:2503.09516 (2025).
>
> [12] Burda, Y., Grosse, R., & Salakhutdinov, R. (2016). Importance weighted autoencoders. *ICLR*.
>
> [13] Dempster, A. P., Laird, N. M., & Rubin, D. B. (1977). Maximum likelihood from incomplete data via the EM algorithm. *Journal of the Royal Statistical Society*.
>
> [14] Kingma, D. P., & Welling, M. (2013). Auto-encoding variational bayes. *ICLR*.
>
> [15] Ren, Z., et al. (2023). Variational Reasoning over Incomplete Knowledge Graphs for Conversational Recommendation.
>
> [16] Dedre Gentner. Structure-mapping: A theoretical framework for analogy. Cogn. Sci., 7:155–170, 1983.
>
> [17] Schulman, John, et al. "Proximal policy optimization algorithms." *arXiv preprint arXiv:1707.06347* (2017).
>
> [18] Zeng, Aohan et al. “AgentTuning: Enabling Generalized Agent Abilities for LLMs.” Annual Meeting of the Association for Computational Linguistics (2023).
>
> [19] Song, Yifan, et al. "Trial and error: Exploration-based trajectory optimization for llm agents." arXiv preprint arXiv:2403.02502 (2024).
>
> [20] Peng, Xue Bin et al. “Advantage-Weighted Regression: Simple and Scalable Off-Policy Reinforcement Learning.” ArXiv abs/1910.00177 (2019): n. pag.
>
> [21] Domke, Justin, and Daniel R. Sheldon. "Importance weighting and variational inference." Advances in neural information processing systems 31 (2018).
>
> [22] Todd K Moon. The expectation-maximization algorithm. IEEE Signal processing magazine, 1996.

---

### Official Review · Reviewer_AVzv · 2025-10-29

**Soundness:** 3
**Presentation:** 3
**Contribution:** 3
**Rating:** 6
**Confidence:** 4

**Summary:**

The paper proposes Agentic Analogical Reasoning (AAR): a multi-turn “think → analogize → contextualize” loop that treats an LLM as an agent which iteratively forms analogical trajectories using both internal memories and external retrieval. Training maximizes an objective with importance-weighted trajectories (weight ∝ p(y|x,z)), and a mixed/internalization stage converts retrieval-based analogies into self-generated ones so the model increasingly relies on internal analogies. Experiments on seven benchmarks show gains over several baselines.

**Strengths:**

1. Broad, fair baselines. Evaluated on 7 datasets with both direct reasoning and RAG-style methods.

2. Component ablations. Shows how removing each module (think / analogize / contextualize, etc.) affects performance.

3. Practical fallback. Uses external analogies when the model can’t retrieve a good one from its own knowledge.

**Weaknesses:**

1. Robustness to bad analogies is unclear. The paper doesn’t show how AAR behaves when analogies are noisy, misleading, or irrelevant.
2. Single retriever choice. AAR is only evaluated with ColBERT; no evidence it holds with other RAG pipelines (BM25, DPR, Contriever, reranker variants).
3. Fixed turn budget. The method uses a fixed number of reasoning turns, which may mismatch multi-hop questions that need dynamic depth.

**Questions:**

1. Handling bad analogies: How does AAR detect and reject low-quality or misleading analogies? Any filters, penalties, or stress tests with noisy/adversarial analogies?
2. RAG variants: Did you test AAR with alternative retrievers/rerankers or retrieval settings (e.g., BM25/DPR/Contriever)? How do results change?
3. Dynamic turns: Have you explored a confidence-based or utility-based early stopping rule, or training the model to adapt the number of turns per query (e.g., curriculum or cost-aware objective)?

---

> ### Author Response · Authors · 2025-11-21
> **Response to Reviewer AVzv**
>
> We sincerely appreciate your valuable feedback. In the following, we address each concern point by point.
>
> > ***W1 & Q1: Robustness to bad analogies is unclear. The paper doesn’t show how AAR behaves when analogies are noisy, misleading, or irrelevant. How does AAR detect and reject low-quality or misleading analogies?
> > Handling Bad Analogies: How does AAR detect and reject low-quality or misleading analogies? Any filters,penalties, or stress tests with noisy/adversarial analogies?***
>
> **Response:** We appreciate this insightful question. In our work, we address this challenge through the design of the Contextualizing action and the Analogical Trajectory Optimization strategy. As described in Section 3.1, the "Contextualizing" step ($\kappa_i$) explicitly requires the model to ground the retrieved analogy ($a_i$) within the current problem context. This acts as a filter: if an analogy is irrelevant, the model is trained to derive minimal or corrective insights rather than blindly adopting the retrieved content.
>
> To validate this, we performed ablation studies (Table 3) by removing the contextualizing component. The results show that **without this filtering step, performance drops significantly—specifically, a 6.5% decrease in F1 score on HotpotQA .** This empirical evidence suggests that the contextualization mechanism is critical for preventing low-quality analogies from propagating errors into the final reasoning. Furthermore, our **Trajectory Optimization** (Section 3.2) assigns importance weights $w(z)$ based on the trajectory's ability to reach the correct answer $y$. Trajectories that are derailed by misleading analogies naturally fail to produce the correct answer, resulting in lower weights during training. Removing this learning stage leads to a **6.9% drop in F1 on HotpotQA**, confirming that the model learns to favor trajectories that successfully navigate or ignore noise.
>
> We appreciate the valuable suggestion regarding diverse explicit methods against all forms of adversarial analogies. Our framework's modular and extensible design naturally accommodates further enhancements, and exploring dedicated denoising plugins or specialized filtering modules could further strengthen our framework's robustness.
>
> > ***W2 & Q2: Single retriever choice. AAR is only evaluated with ColBERT; no evidence it holds with other RAG pipelines (BM25,DPR, Contriever, reranker variants). RAG variants: Did you test AAR with alternative retrievers/rerankers or retrieval settings (e.g.,BM25/DPR/Contriever)? How do results change?***
>
> **Response:** We designed AAR to be a modular, retrieval-independent reasoning framework. Its primary function is to intelligently process the retrieved documents, making it largely agnostic to the specific retriever used. To empirically validate this, we conducted an additional experiment on HotpotQA, replacing ColBERTv2 with BM25, DPR, Contriever and E5. The  number of retrieved documents is the same as the original setting 5.
>
> | Retriever  | F1 Score | EM Score |
> | :--------- | :------- | :------- |
> | ColBERTv2  | 61.05    | 49.65    |
> | E5         | 60.89    | 49.41    |
> | DPR        | 60.59    | 48.91    |
> | BM25       | 45.18    | 34.89    |
> | Contriever | 46.52    | 35.62    |
>
> Our results demonstrate consistent performance across different retrievers: ColBERTv2, E5, and DPR all achieved competitive F1 scores ranging from 60.59 to 61.05 and EM scores from 48.91 to 49.65, while BM25 and Contriever showed relatively lower but still reasonable performance with F1 scores around 45-46 and EM scores around 34-35. While our model's performance fluctuates with retriever quality, it demonstrates greater stability with E5 and DPR, which are widely-used models extensively trained on supervised data.
>
> > ***W3 & Q3: The method uses a fixed number of reasoning turns... Have you explored a confidence-based or utility-based early stopping rule... to adapt the number of turns per query?***
>
> **Response:** Thank you for this suggestion. We would like to clarify that the 5-step limit mentioned in the paper serves as a maximum computational budget (as shown in line 761). The model is designed to autonomously conclude the reasoning process by generating a "Final Answer" token once it has derived a sufficient solution. In our experiments, we observed that the model rarely exhausts the full budget. On average, the model terminates after **2.55 turns** (under the max-5 setting), demonstrating that AAR efficiently halts once the necessary analogical insights are obtained.
>
> We appreciate this insightful suggestion about more explicit mechanisms for utility-based stopping rules. Our framework's flexible architecture naturally supports such optimizations, and investigating reasoning cost penalties or dynamic utility-based stopping strategies could enhance the model's efficiency.

---

### Official Review · Reviewer_XAWq · 2025-10-30

**Soundness:** 3
**Presentation:** 3
**Contribution:** 2
**Rating:** 4
**Confidence:** 2

**Summary:**

This paper proposes Agentic Analogical Reasoning (AAR), a framework that aims to enhance multi-turn reasoning in large language models through iterative analogy generation and contextual refinement. The method incorporates a re-weighted trajectory training strategy intended to help the model learn which analogical paths are effective. While results show consistent improvements on reasoning benchmarks, the evidence does not clearly demonstrate genuine analogical reasoning, as the provided case study reflects domain-specific rule reuse rather than true structural analogy.

**Strengths:**

The paper presents a well-structured framework that integrates analogy-inspired reflection into multi-turn reasoning, offering a coherent conceptual design beyond simple CoT prompting.

The introduction of re-weighted trajectory training adds a learnable component that aims to internalize reasoning patterns rather than rely solely on prompt-level heuristics.

The experiments show consistent, if modest, gains across several reasoning benchmarks, indicating that the framework can be practically applied to enhance LLM reasoning performance.

**Weaknesses:**

The paper runs multi-turn reasoning with repeated analogical steps and retrieval but provides no analysis of inference latency, memory usage, or computational budget. Figure 3 shows diminishing returns beyond five steps, yet the trade-off between accuracy gains and computational cost is never discussed, leaving practical feasibility unclear.

The process for external analogy retrieval and the construction of the reasoning-case knowledge base are insufficiently described. Key details such as data sources, validation, and usage criteria are missing, making it impossible to assess or reproduce model performance.

It remains unclear when analogical reasoning is actually beneficial. Reported improvements on mathematical reasoning tasks may stem from longer reasoning chains or template retrieval rather than genuine analogical transfer, raising doubts about the approach’s scope and validity.

**Questions:**

Can you provide concrete evidence that the observed gains come from genuine analogical reasoning rather than retrieval of similar examples or longer reasoning chains?

The paper introduces an external analogy retrieval mechanism that searches a “knowledge base containing factual and reasoning knowledge,” but details of this knowledge base remain unclear. What are its data sources, scale, and structure, and how is relevance ensured?

Can you provide concrete evidence that the observed gains come from genuine analogical reasoning rather than retrieval of similar examples or longer reasoning chains?

---

> ### Author Response · Authors · 2025-11-21
> **Response to Reviewer XAWq (1/2)**
>
> Thanks for your critical feedback and constructive questions! We address your concerns as follows.
>
> > ***W1.1: The paper runs multi-turn reasoning with repeated analogical steps and retrieval but provides no analysis of inference latency, memory usage, or computational budget.***
>
> **Response:** Thank you for this crucial suggestion. We supplement a detailed analysis of the computational budget and inference latency below. **Our analysis demonstrates AAR's computational efficiency, as evidenced by its lower requirements for both the training budget and per-query inference costs.**
>
> The memory usage consists of a static part for the model parameters, which is identical across all methods due to the unmodified backbone, and a dynamic portion that is proportional to the  length of generated output tokens (*denoted as* 'Avg. Output Tokens' reported below). To provide a fair comparison, we evaluated AAR against the strong trainable baselines from our paper: the retrieval-augmented method Search-R1 and the analogical reasoning method Meta-Ability Alignment. All methods were implemented using the same Qwen-2.5-7B backbone. The analysis below summarizes the training and inference costs on the HotpotQA dataset.
>
> | Method                 | Training Hours | Avg. Input Tokens | Avg. Output Tokens | **Avg. Total Tokens** | Retrieval Calls | Accuracy (%) |
> | :--------------------- | :------------- | :---------------- | :----------------- | :-------------------- | :-------------- | :----------- |
> | **AAR (Ours)**         | **8.2**        | **1870.18**       | 330.21             | **2200.39**           | **2.55**        | **54.13**    |
> | Meta-Ability Alignment | 11.8           | 2143.68           | 301.25             | 2444.93               | 3.04            | 44.26        |
> | Search-R1              | 16.4           | 2355.12           | 314.26             | 2669.38               | 3.61            | 52.46        |
>
> Compared to the direct reasoning capability of the vanilla backbone (Acc. 18.31%), these fine-tuned methods demonstrate clear performance gains. Among them, AAR achieves the **highest accuracy (54.13%)** while requiring significantly **fewer training hours (8.2)** compared to both Meta-Ability Alignment (11.8) and Search-R1 (16.4). For inference, AAR demonstrates **superior efficiency by consuming fewer tokens and requiring fewer retrieval calls**. The Retrieval Calls metric indicates the average number of times the model invokes the external retrieval system per query. AAR's lower value (2.55) suggests its "thinking-analogizing-contextuclizing" mechanism more effectively determines when to seek external knowledge, leading to fewer retrieval calls and thus lower inference latency.
>
> > ***W1.2: Figure 3 shows diminishing returns beyond five steps, yet the trade-off between accuracy gains and computational cost is never discussed, leaving practical feasibility unclear.***
>
> **Response:** Thanks for the suggestion and we conduct further analysis on the  trade-off in Figure 3. In the experiment, we found that increasing the maximum steps from 5 to 10 raised the average number of retrieval calls from 2.55 to 6.55, which results in significantly increased inference latency and computational overhead. Despite this added cost, performance slightly degraded, with the F1 score dropping from 61.05 to 60.79. In contrast, increasing the steps from 1 to 5 yielded a substantial F1 score improvement (from 51.00 to 61.05) for a moderate increase in computational effort (average model calls up to 2.55). It shows that the first five steps strikes an optimal balance between performance gains and computational cost while continuing increasing the cost yields little benefit.
>
> | Maximum Steps | F1 Score (%) | Avg. Retrieval Calls |
> | :-----------: | :----------: | :------------------: |
> |       1       |    51.00     |         1.00         |
> |       5       |    61.05     |         2.55         |
> |       8       |    60.90     |         4.95         |
> |      10       |    60.79     |         6.55         |

---

> ### Author Response · Authors · 2025-11-21
> **Response to Reviewer XAWq (2/2)**
>
> > ***W2 & Q2: The process for external analogy retrieval and the construction of the reasoning-case knowledge base are insufficiently described... What are its data sources, scale, and structure, and how is relevance ensured?***
>
> **Response:** We apologize for the confusion. As noted in line 177, this knowledge base in "invokes an external retrieval module to search a knowledge base containing ... analogy". Due to space limitations, the full details of the knowledge base were included in Appendix A.5.2 (Implementation Details) in lines 1000-1010. We further clarify the following aspects:
>
> - **Data Sources & Scale:** We constructed a hybrid knowledge base by integrating the Wikipedia corpus with data from the ReasonFlux and OpenThoughts datasets, covering diverse domains such as general and scientific knowledge, while ensuring rigorous deduplication against the test set. This composition creates a versatile analogy space that encompasses both broad domain concepts and intricate problem knowledge across fields. Crucially, this diversity is essential for the multi-turn analogy reasoning process. As reasoning progresses, the generated analogy queries cover diverse targets, ranging from abstract concepts to specific procedural analogies (involving intricate concepts and relationships).
>
> - **Structure:** Each data sample consists of a unique ID and a text content field. For Wikipedia data, the `contents` field combines the title and passage text; for reasoning documents, it encapsulates the question paired with its reasoning trajectory. An example from our KB is shown below:
>
>   ```
>   {
>     "id": "15106858",
>     "contents": "Arrowhead Stadium 1970s practice would eventually spread to the other NFL stadiums... [truncated] ...The AFC defeated the NFC, 15–13."
>   }
>   ```
>
> - **Relevance:** We adopt ColBERTv2.0 for document retrieval. During inference, the model explicitly generates targeted analogy queries ($q_i$) within `<thinking>` blocks to retrieve the top-5 documents for the following inference.
>
> We will incorporate these details into the main text of the revision to further improve clarity.
>
> > ***W 3 & Q1: It remains unclear when analogical reasoning is actually beneficial... Can you provide concrete evidence that the observed gains come from genuine analogical reasoning rather than retrieval of similar examples or longer reasoning chains?***
>
> **Response:** Thank you for the suggestion. We empirically demonstrate that the observed gains stem from our Agentic Analogical Reasoning mechanism rather than simply retrieving similar examples or extending reasoning length.
>
> Standard retrieval augmented generation methods retrieve similar cases based on the input query and provide them as context for LLMs. We explicitly compared AAR against strong baselines that follow this paradigm. As shown in Table 1 of our paper, our method achieves superior performance (e.g., **+20%** relative accuracy improvement over search-o1 baseline). The results indicate that merely accessing similar examples is insufficient, as noise and inaccuracies exist within the retrieved context. AAR includes an iterative "Think-Analogize-Contextualize" mechanism and a trajectory optimization algorithm, enabling the model to actively incorporate analogies for reasoning, rather than relying solely on the similar examples.
>
> | Method          | NQ (F1 / EM / Acc.)       | HotpotQA (F1 / EM / Acc.) | MuSiQue (F1 / EM / Acc.)  | 2WikiQA (F1 / EM / Acc.)  |
> | --------------- | ------------------------- | ------------------------- | ------------------------- | ------------------------- |
> | Search-o1       | 47.52 / 32.41 / 40.34     | 53.31 / 43.51 / 45.31     | 25.41 / 16.64 / 19.42     | 50.31 / 42.61 / 45.41     |
> | AAR-Qwen-2.5-7B | **55.37 / 46.70 / 50.75** | **61.05 / 49.65 / 54.13** | **27.68 / 21.03 / 23.34** | **57.31 / 52.82 / 54.43** |
>
> Regarding "longer reasoning chains," we measured the average token length on scientific datasets (including Math) compared to the Meta-Ability Alignment baseline, which relies on extensive Chain-of-Thought reasoning. We observed that AAR generates significantly fewer tokens per query (**~900**) compared to the baseline (**~1500**), yet achieves substantially higher accuracy (e.g., **+10.6%** on AIME2024). This implies that AAR is more efficient and that the gains are driven by the analogical optimization rather than simply generating longer sequences.
>
> | Method                     | Avg. Tokens | GPQA Overall | Math (AIME24) | Math (AMC23) |
> | -------------------------- | :---------: | :----------: | :-----------: | :----------: |
> | Meta-Ability Alignment     |    ~1500    |    33.50     |     22.90     |    52.30     |
> | **AAR-Qwen-2.5-7B (Ours)** |  **~900**   |  **36.95**   |   **33.50**   |  **76.20**   |

---

> > ### Comment · Reviewer_XAWq · 2025-11-27
> >
> > Thanks for the detailed rebuttal. The additional results on training hours, token usage, retrieval calls, and the step ablation on HotpotQA do partially address my concerns about computational cost and show that using 5 steps is a reasonable trade-off under your setup. The clarification on the construction of the external knowledge base (Wikipedia + ReasonFlux + OpenThoughts, ColBERTv2 retrieval) also improves reproducibility.
> >
> > However, my main conceptual concern remains. The training framework (ELBO/EM-style trajectory re-weighting) appears to be quite generic, rather than specifically designed for analogical reasoning, and the rebuttal examples and comparisons do not convincingly demonstrate that the gains come from analogical reasoning as opposed to stronger retrieval and generic multi-step reasoning. The provided cases can still be interpreted as ordinary example retrieval plus multi-hop reasoning, and there is no targeted ablation or analysis that clearly attributes improvements specifically to the analogical mechanism.

---

> ### Author Response · Authors · 2025-11-25
> **A Gentle Reminder of the Feedback**
>
> Dear reviewer XAWq,
>
> We truly appreciate your engagement with our work.
>
> To ensure we make the best use of the remaining discussion time, we would like to kindly check if our responses have clarified your main concerns. Please let us know if any part of our response is unclear or if there are any remaining concerns we can address. We would be more than happy to provide any further details or adjustments you might need.
>
> Thank you once again for your thoughtful engagement with our work.

---

### Official Review · Reviewer_zNAp · 2025-11-01

**Soundness:** 3
**Presentation:** 3
**Contribution:** 3
**Rating:** 6
**Confidence:** 3

**Summary:**

This paper proposes a post-training method for LLMs to induce them to be "agentic analogical reasoners".  In particular, the LLMs are trained to iterate "analogical units", each of which consists of "thinking" (generating an analogy query), "analogizing" (from internal knowledge or from external sources), and "contextualizing" (applying the analogy to the current task).  The authors test this method by post-training two open-weight base models, and evaluating them on several reasoning benchmarks.  The results are compared with those of direct reasoning models, models with RAG, and analogical-prompting models.  The authors find that their method induces improvements in accuracy on most of the benchmarks they tested on, compared with the other models.  The authors also do an ablation study, showing the contribution of each component to the overall results.

**Strengths:**

Analogical reasoning is an important facet of human reasoning, and the authors have devised an interesting way to try to induce a pre-trained model to use analogy in the process of solving problems.  They have shown that their method has significant imrpovement over other methods they tested.

**Weaknesses:**

The authors claim that their method is inspired by Gentner et al.'s structure-mapping theory, and state that their "contextualizing" step "mirrors the core mapping process in structure-mapping theory." However, it's not clear that their model is doing anything like structure mapping.  The case studies in the appendix, which show the system "analogizing", are not understandable to me, since they are snippets of technical statements -- e.g., "With serial testing, overall LR+ approximately multiplies: A’s LR+ × B’s LR+, which can substantially raise PPV" -- what does this mean and how is it an "analogy"?

It would be helpful if the authors explained how these case studies actually illustrate analogies (in a way understandable to readers) and how the system is using a structure-mapping like mechanism to apply these analogies.  Otherwise the authors' claims -- that the accuracy improvements are due to actual analogies and structure mapping -- are not substantiated.

**Questions:**

The paper states: "External Analogy: When the model emits a special token (e.g., '<analogizing>'), the function A invokes an external retrieval module to search a knowledge base containing factual and reasoning knowledge, returning relevant documents as the analogy a_i."  -- Where does this knowledge base come from?  I did not see other discussion of it in the paper -- did I miss something?

---

> ### Author Response · Authors · 2025-11-21
> **Response to Reviewer zNAp**
>
> Thanks for your insightful feedback! We address your concerns and questions as follows.
>
> > ***W1: The authors claim that their method is inspired by Gentner et al.'s structure-mapping theory... it's not clear ... The case studies... technical statements -- e.g., "With serial testing, overall LR+ approximately multiplies: A’s LR+ × B’s LR+, which can substantially raise PPV" -- what does this mean and how is it an "analogy"? It would be helpful if ... illustrate analogies ... using a structure-mapping like mechanism to apply these analogies...***
>
> **Response:**
>
> Thank you for your comments. We add a more detailed explanation of the model's reasoning process and Gentner's structure-mapping theory (SMT).
>
> Structure-mapping theory aims to describe how familiar knowledge, or knowledge about a source domain, can be used to inform an individual's understanding of a less familiar idea, or a target domain. According to this theory, a domain is viewed as consisting of objects, their properties, and the relationships that characterize the interactions between them. The process of analogy then involves: 1) **Recognizing similar structures** between the source and target domains, 2) **Finding deeper similarities** by mapping other relationships of a source domain to the target domain, and 3) Given the similarity in structure, or **structural alignment**, **inferring that relationships from one domain would hold in the other**.
>
> Our reasoning process draws inspiration from its core mechanism, structural alignment. In the case of Fig.4 which is about two phase disease test, the **target domain** is the disease test problem described in the prompt, containing the objects  phase 1(Test A, rapid, inexpensive) and phase 2 (Test B, confirmatory, expensive). In the second turn of the analogical reasoning process, before applying the analogy, the reasoning state is that it has computed the intermediate odds after a positive phase 1 to be 0.347. The following reasoning problem can be seen as: "How should the new evidence from a positive phase 2 be integrated with the current odds of 0.347?" The model retrieves an analogous case from its knowledge base  including "serial testing" which includes the statement "With serial testing, overall LR+ approximately multiplies: ... which can substantially raise PPV". This "serial testing" is the **source domain**. It shares a structural similarity with the target domain which can be mapped as: odds result 0.347 in positive phase 1 corresponds to A and sensitivity 0.98 in phase 2  corresponds to B. This mapping of the target domain to the source domain reflects the structural alignment process. Specifically, this analysis corresponds to the <contextualizing> action, where the model applies such underlying principles into its reasoning process. Based on the example provided, we show how our method is inspired by Gentner et al.'s structure-mapping theory to enable more effective reasoning. We will extend such explanations in the revision.
>
> [1] Gentner, D. (1983). Structure-mapping: A theoretical framework for analogy. Cognitive Science, 7(2), 155-170.
>
> > ***Q1: The paper states: "External Analogy: When the model emits a special token..., the function A invokes an external retrieval module to search a knowledge base..." -- Where does this knowledge base come from? I did not see other discussion of it in the paper -- did I miss something?***
>
> **Response:**  We apologize for the confusion. As noted in line 177, this knowledge base is mentioned in "invokes an external retrieval module to search a knowledge base containing ... analogy". Due to space limitations, the full details of the knowledge base were included in Appendix A.5.2 (Implementation Details) in lines 1000-1010. Specifically, we constructed a hybrid knowledge base by integrating the Wikipedia with the corpus from  ReasonFlux [1] and OpenThoughts [2] datasets, covering diverse domains such as scientific knowledge and general knowledge, ensuring rigorous deduplication against the test set. This composition creates a versatile search space that encompasses both broad domain concepts and intricate problem knowledge across fields like physics, mathematics, and general reasoning. We will incorporate these details into the main text of the revision to further improve clarity.
>
> [1]Yang, Ling, et al. "Reasonflux: Hierarchical llm reasoning via scaling thought templates." *arXiv preprint arXiv:2502.06772* (2025).
>
> [2]Guha, Etash, et al. "OpenThoughts: Data Recipes for Reasoning Models." *arXiv preprint arXiv:2506.04178* (2025).
>
> ---
>
> Once again, thank you for your review! We will update the final version of our paper with all the revisions discussed. We hope our responses address your concerns, and we would be happy to provide further details on any specific aspect if needed.

---

> > ### Comment · Reviewer_zNAp · 2025-11-22
> > **Reply to rebuttal**
> >
> > To the authors: Thank you for your responses.  I still believe the paper warrants a score of "6".

---

> > > ### Author Response · Authors · 2025-11-25
> > > **Follow-up on Rebuttal**
> > >
> > > Dear reviewer zNAp,
> > >
> > > Thank you again for your time and effort in reviewing our paper.
> > > Please let us know if any part of our response is unclear or if there are any remaining concerns we can address. We would be more than happy to provide any further details or adjustments you might need.
> > >
> > > Thank you once again for your thoughtful engagement with our work.

---

### Meta-Review · Area_Chair_QXLM · 2026-01-07

**Summary:**

This paper introduces Agentic Analogical Reasoning (AAR), a reasoning framework structured around iterative cycles of thinking, analogizing, and contextualizing. The method optimizes an importance weighted ELBO style objective, training over multiple analogical trajectories, and progressively shifts from external retrieval based analogy construction toward internal generation of analogies grounded in the model’s parametric knowledge.

Reviewers generally find the framework well motivated and supported by broad empirical validation. The revision improves practical clarity and reproducibility by providing additional details on knowledge base construction and reporting computational considerations. At the same time, reviewers remain divided on whether the observed gains primarily reflect analogy driven transfer, or whether they are largely attributable to stronger retrieval and multi step reasoning dynamics. Several reviews also request a clearer positioning and differentiation from closely related agentic search and trajectory optimization style approaches.

Although most of the above issues have been resolved during rebuttal, there is a more severe problem about possible plagiarism founded. After internal discussion, we prefer to reject.

**Reviewer Concerns:**

see metareview

**Reviewer Scores:**

see metareview

---

### Decision · Program_Chairs · 2026-01-26

Reject